# Saturated very long chain fatty acid configures glycosphingolipid for lysosome homeostasis in long-lived *C. elegans*

Feng Wang[1,4], Yuxi Dai[1,4], Xufeng Zhu[2], Qilong Chen[3], Huanhu Zhu [2], Ben Zhou[3], Haiqing Tang[1✉] & Shanshan Pang [1✉]

The contents of numerous membrane lipids change upon ageing. However, it is unknown whether and how any of these changes are causally linked to lifespan regulation. Acyl chains contribute to the functional specificity of membrane lipids. In this study, working with *C. elegans*, we identified an acyl chain-specific sphingolipid, C22 glucosylceramide, as a longevity metabolite. Germline deficiency, a conserved lifespan-extending paradigm, induces somatic expression of the fatty acid elongase ELO-3, and behenic acid (22:0) generated by ELO-3 is incorporated into glucosylceramide for lifespan regulation. Mechanistically, C22 gluco-sylceramide is required for the membrane localization of clathrin, a protein that regulates membrane budding. The reduction in C22 glucosylceramide impairs the clathrin-dependent autophagic lysosome reformation, which subsequently leads to TOR activation and longevity suppression. These findings reveal a mechanistic link between membrane lipids and ageing and suggest a model of lifespan regulation by fatty acid-mediated membrane configuration.

[1] School of Life Sciences, Chongqing University, Chongqing, China. [2] School of Life Science and Technology, ShanghaiTech University, Shanghai, China. [3] CAS Key Laboratory of Nutrition, Metabolism and Food Safety, Shanghai Institute of Nutrition and Health, University of Chinese Academy of Sciences, Chinese Academy of Sciences, Shanghai, China. [4] These authors contributed equally: Feng Wang, Yuxi Dai. ✉email: hqtang@cqu.edu.cn; sspang@cqu.edu.cn

Plasma and organelle membranes are composed of thousands of lipid species, which have function in addition to their roles as structural components. Numerous studies have linked the complexity of membrane lipids to signal transduction, cellular function, and animal physiology[1]. Recently, the central roles of membrane lipids in the animal aging process have begun to emerge. The contents and composition of numerous membrane lipids correlate with the aging process. For example, the levels of many sphingolipids and related metabolites are increased in long-lived humans and organisms[2], implying their crucial roles in the longevity response. However, the mechanistic link between membrane lipids and aging is largely unknown.

Fatty acids play important roles in regulating membrane lipid function. For example, the acyl (fatty acid) chains of sphingolipids confer specificity to sphingolipid-dependent signaling. On the membrane, sphingolipids interact with cholesterol to form spatially compartmentalized microdomains, referred to as lipid rafts, which control the structure, localization, and activity of raft-associated proteins[3]. Sphingolipid acyl chains typically contain from 14 to 26 carbon atoms, which determine the raft protein(s) associated with a sphingolipid and thereby control the specificity of downstream signaling. In line with this, the crucial roles of acyl chain-specific sphingolipids and their respective molecular mechanisms have been uncovered in diverse biological processes[4,5]. Therefore, identifying acyl chain-specific membrane lipids that affect the aging process would lay the groundwork for mechanistic studies linking membrane components to aging.

Fatty acids have been found to play crucial roles in longevity regulation. Studies with model organisms such as *C. elegans* have revealed that several fatty acid species exert lifespan-extending effects[2], namely, monounsaturated oleic acid[6], palmitoleic acid, cis-vaccenic acid[7], and oleoylethanolamide[8], as well as polyunsaturated α-linolenic acid[9], arachidonic acid, and dihomo-g-linolenic acid[10]. Fatty acids can act as signaling molecules to directly regulate lifespan[8]; however, in most cases, their mechanisms of action are unknown. It is conceivable that some fatty acids may modulate the aging process through their configuration of membrane lipids.

Germline-deficient *C. elegans* is an established longevity model with rewired lipid metabolism[11,12]. The lifespan-extending effects of germline removal might be evolutionarily conserved, as loss of germline stem cells increases the lifespan of *Drosophila melanogaster*[13], and castration increases longevity in men[14]. A hallmark of germline-deficient *C. elegans* is metabolic reprogramming. Lipids that were originally assigned to the germline, accumulate in the soma accompanied with reprogrammed fatty acid metabolism[15–17]. Importantly, these accumulated lipids function not only as energy reserves for somatic maintenance but also act as signaling molecules to regulate the conserved longevity transcription factor SKN-1/Nrf and thus retard the aging process[6,15]. Moreover, several fatty acid metabolic enzymes, including fatty acid desaturase FAT-6[6,15], acyl-CoA synthetases[17], diacylglycerol acyltransferase DGAT-2[18], triglyceride lipases LIPL-4[16] and LIPS-17[19], and fatty acyl reductase FARD-1[19], are essential for lifespan extension in *C. elegans* that lack the germline. These studies suggest tight links between reproduction, lipid metabolism, and the aging process; however, the lipid-based signaling through which the germline influences somatic lifespan is not well understood.

The composition of fatty acids is controlled by a series of fatty acid elongases and desaturases, which determine the lengths of carbon chains and the degrees of unsaturation, respectively. The function of fatty acid desaturation in germline-mediated longevity has been explored, with Δ9 desaturase FAT-6 identified as a regulator of lifespan in *C. elegans*[6,15]. Several fatty acid elongases have also been found to be upregulated upon germline loss in *C.*

*elegans*[15,17]. However, whether fatty acid elongation is involved in the longevity response is unknown. We therefore screened fatty acid elongases in germline-deficient *C. elegans* and identified the previously uncharacterized fatty acid elongase ELO-3 as a critical regulator of longevity through SKN-1 activation. Further analysis identified behenic acid (BA, 22:0) as the product of ELO-3, which is a saturated very-long-chain fatty acid (SVLCFA) that regulates lifespan through incorporation into glucosylceramide. This acyl chain-specific sphingolipid functions through the membrane protein clathrin to modulate downstream lysosome homeostasis, TOR signaling, and SKN-1 activation. We propose that the lipid signaling uncovered here may affect longevity in other species, including mammals.

## Results

### The fatty acid elongase ELO-3 regulates SKN-1 activation upon germline loss.

Fatty acid elongases (ELOs) catalyze the rate-limiting and substrate-specific step of fatty acid elongation through which acyl-CoA and malonyl-CoA are condensed to generate 3-ketoacyl-CoA (Supplementary Fig. 1a). To date, nine *elo* genes have been discovered in the *C. elegans* genome, and the metabolic functions of ELO-1/2/5/6 have been well characterized[20]. In this study, we explored the roles of ELOs in longevity using germline-deficient *glp-1* mutants[11]. The transcription factor SKN-1 responds to lipid changes in *glp-1* mutants[15]; therefore, we knocked down each *elo* gene and examined the expression of *gst-4p*::GFP[21], an established reporter of SKN-1 transcriptional activity. The results showed that *elo-3* RNAi significantly suppressed the induction of *gst-4p*::GFP (Fig. 1a). Quantitative PCR analysis confirmed the suppressive effects of *elo-3* RNAi on other SKN-1 target genes (Fig. 1b). Importantly, germline depletion led to upregulated *elo-3* mRNA levels (Fig. 1c), indicating the activation of ELO-3 in response to germline loss.

Upon activation, the transcription factor SKN-1 is translocated from the cytoplasm to the nucleus. Germline removal greatly increased the nuclear occupancy of SKN-1, as previously reported[15] (Fig. 1d, e). As expected, *elo-3* RNAi treatment almost completely abrogated SKN-1 nuclear accumulation (Fig. 1d, e), while the protein levels of SKN-1 were unaffected (Supplementary Fig. 1b), suggesting that ELO-3 activates SKN-1 by increasing its nuclear translocation.

We next sought to determine whether ELO-3 regulates DAF-16 and HSF-1, two essential transcription factors that modulate the longevity of *glp-1* mutants. However, the expression of the DAF-16 reporter *sod-3p*::GFP[22] and the HSF-1 reporter *hsp-16.2p*::GFP[23] was not affected by *elo-3* RNAi (Supplementary Fig. 1c, d), suggesting that ELO-3 is specific for SKN-1 activation in germline-deficient animals.

### ELO-3 regulates stress resistance and longevity in germline-deficient animals.

SKN-1 is critical for the enhanced oxidative stress resistance and longevity of germline-deficient animals[15]. Consistently, the enhanced resistance to tert-butyl hydroperoxide (TBHP), a chemical inducer of oxidative stress, was largely abolished by *elo-3* knockdown in *glp-1* mutants (Fig. 1f). Moreover, the extended lifespan of *glp-1* mutants was dramatically abrogated by *elo-3* RNAi (Fig. 1g), whereas the lifespan of wild-type worms was not affected (Fig. 1h). We also evaluated the involvement of ELO-3 in other longevity models requiring SKN-1, including the insulin/IGF-1 receptor-deficient model with *daf-2* mutation and the dietary-restriction model with *eat-2* mutation, and found that knocking down *elo-3* suppressed the lifespan of the *daf-2* (19% reduction) and *eat-2* (15% reduction) mutants, although to a lesser extent than observed in the germline-deficient

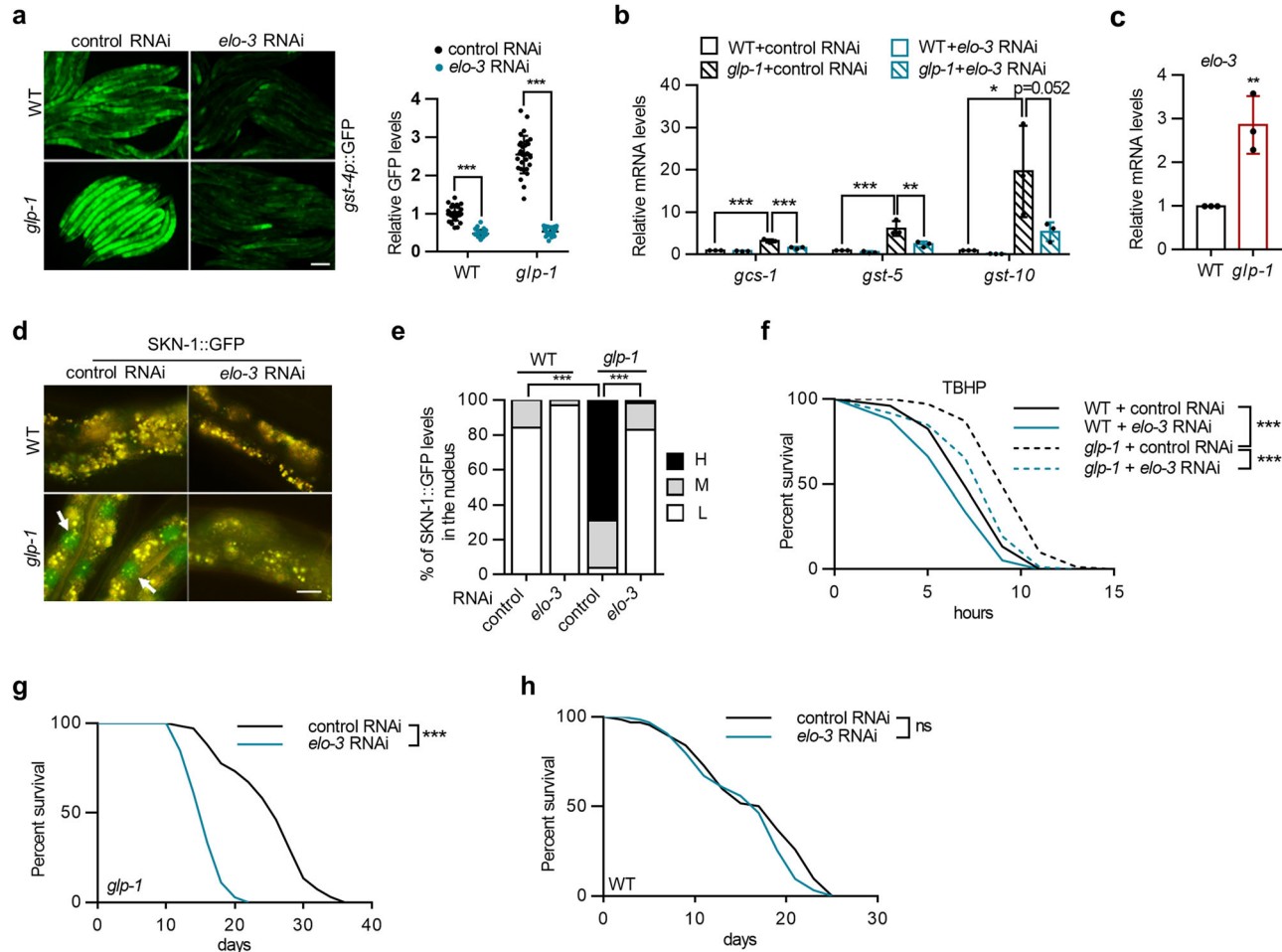

**Fig. 1 ELO-3 regulates SKN-1 activation and longevity in response to germline loss. a** Effects of *elo-3* RNAi treatment on the expression of *gst-4p*::GFP in wild-type and *glp-1* mutant worms. $n = 30$ animals. **b** Effects of *elo-3* RNAi treatment on the expression of SKN-1 target genes. $n = 3$ independent experiments. **c** The mRNA levels of *elo-3* in *glp-1* mutants. $n = 3$ independent experiments. **d** Effects of *elo-3* RNAi treatment on the nuclear accumulation of SKN-1::GFP in the intestine of wild-type and *glp-1* mutant worms. Arrows indicate the nuclear occupancy of SKN-1::GFP in the intestine. **e** Quantification of the nuclear occupancy of SKN-1::GFP. $n = 89/90/99/102$ animals for WT (control RNAi)/WT(*elo-3* RNAi)/*glp-1*(control RNAi)/*glp-1*(*elo-3* RNAi). **f** Effects of *elo-3* RNAi treatment on TBHP resistance in wild-type and *glp-1* mutant worms. **g, h** Effects of *elo-3* RNAi treatment on the lifespan of *glp-1* mutants (**g**) and wild-type controls (**h**). Data are represented as mean ± SD. *$p < 0.05$, **$p < 0.01$, ***$p < 0.001$. **a** was analyzed by two-way ANOVA with Turkey's multiple-comparison test (***$p < 0.001$). **b** was analyzed by one-way ANOVA with Turkey's multiple-comparison test (***$p < 0.001$, **$p = 0.0055$, *$p = 0.0137$). **c** was analyzed by unpaired two-tailed $t$ test ($p = 0.0082$). **e** was analyzed by Chi-square and Fisher's exact test (***$p < 0.001$). See Supplementary Table 1 and 2 for statistical analysis and additional repeats of the survival assays (**g**, **h**) that were analyzed by log-rank (Mantel–Cox) test. Scale bar $= 100 \, \mu m$ for panel **a** and 12.5 $\mu m$ for panel d. Source data are provided as a Source Data file.

animals (37% reduction) (Supplementary Fig. 1e, f). Together, we conclude that ELO-3 is a potent regulator of the stress response and longevity of animals lacking a germline.

**ELO-3 regulates SKN-1 and longevity through the generation of SVLCFA.** ELO-3 was found to regulate cell polarity in *C. elegans*[24]; however, its metabolic function remains unknown. To explore its potential mechanisms in longevity, we constructed a translational reporter for ELO-3 that was mainly expressed in the intestine (Fig. 2a), which is the fat tissue of *C. elegans*, consistent with the role of ELO-3 as a fatty acid elongase.

We expected to find that ELO-3 is critical for the generation of specific fatty acids. We next performed GC–MS/MS analysis and found that most fatty acid species were not significantly affected by *elo-3* knockdown (Supplementary Fig. 2a). However, *elo-3* RNAi dramatically decreased the contents of saturated fatty acids with chains longer than twenty carbon atoms (21:0, 22:0, 23:0,

and 24:0) (Fig. 2b), suggesting that ELO-3 may catalyze the production of these SVLCFA species. Furthermore, we analyzed the fatty acid composition of triglycerides and confirmed that *elo-3* RNAi treatment led to significant reduction in SVLCFA-containing triglycerides (Fig. 2c), while the total triglyceride contents remained unchanged (Supplementary Fig. 2b).

We speculated that one particular SVLCFA might account for the regulation of SKN-1. To explore this possibility, we added exogenous heneicosylic acid (HA) (21:0) and BA (22:0) to *elo-3* RNAi-treated *glp-1* worms, which efficiently restored the corresponding fatty acid levels (Supplementary Fig. 2c), and we found that BA, but not HA, led to the largely reestablished expression of *gst-4p*::GFP (Fig. 2d and Supplementary Fig. 2d) and resistance to TBHP (Fig. 2e). Because lignoceric acid (24:0) is insoluble in DMSO, ethanol, or aqueous solution, its dietary addition is difficult; therefore, we conclude that even-numbered SVLCFAs, likely BA, regulate SKN-1 activation in germline-deficient animals. A previous study reported that oleic acid (OA)

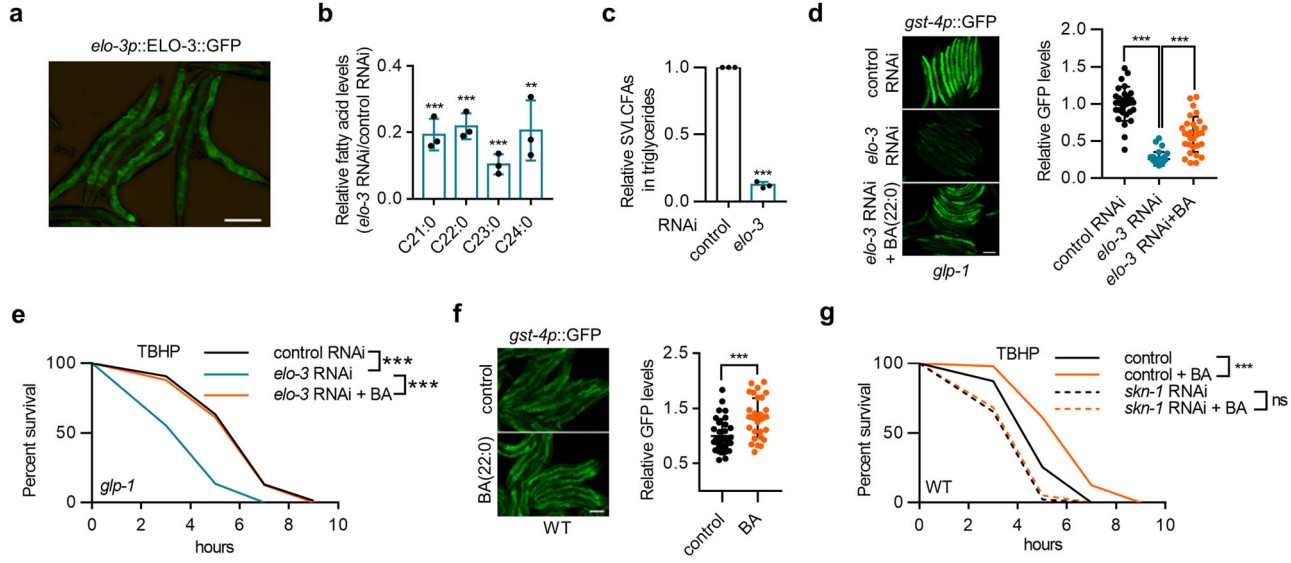

**Fig. 2 ELO-3 regulates SKN-1 activity through BA. a** The expression pattern of *elo-3p*::ELO-3::GFP. **b** Effects of *elo-3* RNAi treatment on the contents of major SVLCFA species in wild-type worms. $n = 3$ independent experiments. **c** Effects of *elo-3* RNAi treatment on the contents of total SVLCFAs in triglycerides. $n = 3$ independent experiments. **d, e** Effects of BA (22:0) supplementation on *gst-4p*::GFP expression **d** ($n = 30$ animals) and TBHP resistance (**e**) in *glp-1* mutants treated with *elo-3* RNAi. **f** Effects of BA supplementation on *gst-4p*::GFP expression in wild-type worms. $n = 32$ animals. **g** *skn-1* RNAi abrogates the effects of BA supplementation on TBHP resistance in wild-type animals. Data are represented as mean ± SD. **$p < 0.01$, ***$p < 0.001$. **b** was analyzed by multiple *t*-test with correction for multiple comparisons using the Holm–Sidak method (***$p < 0.001$, **$p = 0.0033$). **c** and **f** were analyzed by unpaired two-tailed *t*-test (***$p < 0.001$). **d** was analyzed by one-way ANOVA with Turkey's multiple-comparison test (***$p < 0.001$). See Supplementary Table 1 for statistical analysis and additional repeats of the survival assays (**e**, **g**) that were analyzed by log-rank (Mantel–Cox) test. Scale bar = 100 μm for panels **a**, **d**, and **f**. Source data are provided as a Source Data file.

is required for SKN-1 activation in germline-deficient *C. elegans*[6]. However, OA supplementation failed to rescue the expression of *gst-4p*::GFP upon *elo-3* knockdown (Supplementary Fig. 2e), suggesting that ELO-3 does not regulate SKN-1 via OA production.

We next added BA to germline-intact animals and found that dietary BA was able to enhance the expression of *gst-4p*::GFP (Fig. 2f) and SKN-1 target genes (Supplementary Fig. 2f). Consistently, BA supplementation enhanced oxidative stress resistance in a SKN-1-dependent manner (Fig. 2g). Moreover, ELO-3 overexpression also improved oxidative stress resistance in wild-type animals (Supplementary Fig. 2g), suggesting that ELO-3-produced BA is sufficient to exert beneficial effects on SKN-1 activation and the stress response.

**Ceramide biosynthetic enzymes regulate SKN-1 in germline-deficient animals.** How does BA regulate SKN-1? SVLCFAs typically do not function as free fatty acids; they typically act as components of membrane lipids, such as sphingolipids. The acyl chain composition of sphingolipids is crucial for cellular homeostasis. Disruption of sphingolipid composition has been linked to many cellular dysfunctions and human diseases[25]. Intriguingly, several sphingolipid biosynthetic enzymes are involved in the lifespan regulation of *C. elegans*, with both lifespan-extending and -shortening effects observed, possibly due to opposite functions of sphingolipids containing different acyl chains[26,27]. We speculated that certain BA-containing sphingolipids might account for the SKN-1 activation and lifespan extension of germline-deficient animals.

The backbone of a sphingolipid is ceramide, composed of a sphingoid base and a fatty acid chain. The first and rate-limiting step in *C. elegans* ceramide biosynthesis is catalyzed by the serine palmitoyltransferase SPTL-1 (Supplementary Fig. 3a). Therefore, we first examined the role of SPTL-1 in SKN-1 activation. The results showed that *sptl-1* RNAi significantly inhibited the

expression of *gst-4p*::GFP (Fig. 3a) and SKN-1 target genes (Fig. 3b) and reduced the nuclear occupancy of SKN-1::GFP (Fig. 3c) in *glp-1* mutants. Consistently, the enhanced oxidative stress resistance and longevity of *glp-1* mutants were largely abolished by *sptl-1* RNAi (Supplementary Fig. 3b, c). Notably, the lifespan of the wild-type animals was also reduced significantly by *sptl-1* RNAi (Supplementary Fig. 3c), suggesting that global inhibition of sphingolipid production may cause general sickness. In addition, the mRNA levels of *sptl-1* were significantly elevated in animals lacking germline (Supplementary Fig. 3d). Together, these data suggest that the induction of *sptl-1* in response to germline removal may contribute to SKN-1 activation and lifespan extension.

In *C. elegans*, the sphingoid base of ceramide is derived from monomethyl branched-chain fatty acids (mmBCFAs) (Supplementary Fig. 3a)[28,29]. If ceramides were indeed required for SKN-1 activation, then the disruption of mmBCFA biosynthesis would suppress SKN-1 activity. mmBCFAs are generated by the fatty acid elongase ELO-5 and the 3-ketoacyl-CoA reductase LET-767 in *C. elegans*[30,31]. Because RNAi targeting *let-767* and *elo-5* resulted in developmental arrest[30], we diluted *let-767* and *elo-5* RNAi to ensure normal development of *C. elegans* into adulthood. Remarkably, the induction of *gst-4p*::GFP and SKN-1 nuclear accumulation were largely abrogated by *let-767* or *elo-5* RNAi in *glp-1* mutants (Fig. 3d, supplementary Fig. 3e), supporting a critical role for mmBCFA biogenesis in SKN-1 activation. The mRNA levels of *let-767* were also upregulated in *glp-1* mutants (Supplementary Fig. 3d). Collectively, these data suggest that the induction of ceramide biosynthetic enzymes triggers SKN-1 activation in response to germ cell absence.

**C22 ceramide promotes SKN-1 activation and extends lifespan.** Next, we investigated whether the induction of ceramide biosynthetic enzymes activated SKN-1 through ceramide production. Since the ceramide content is balanced by synthesis and

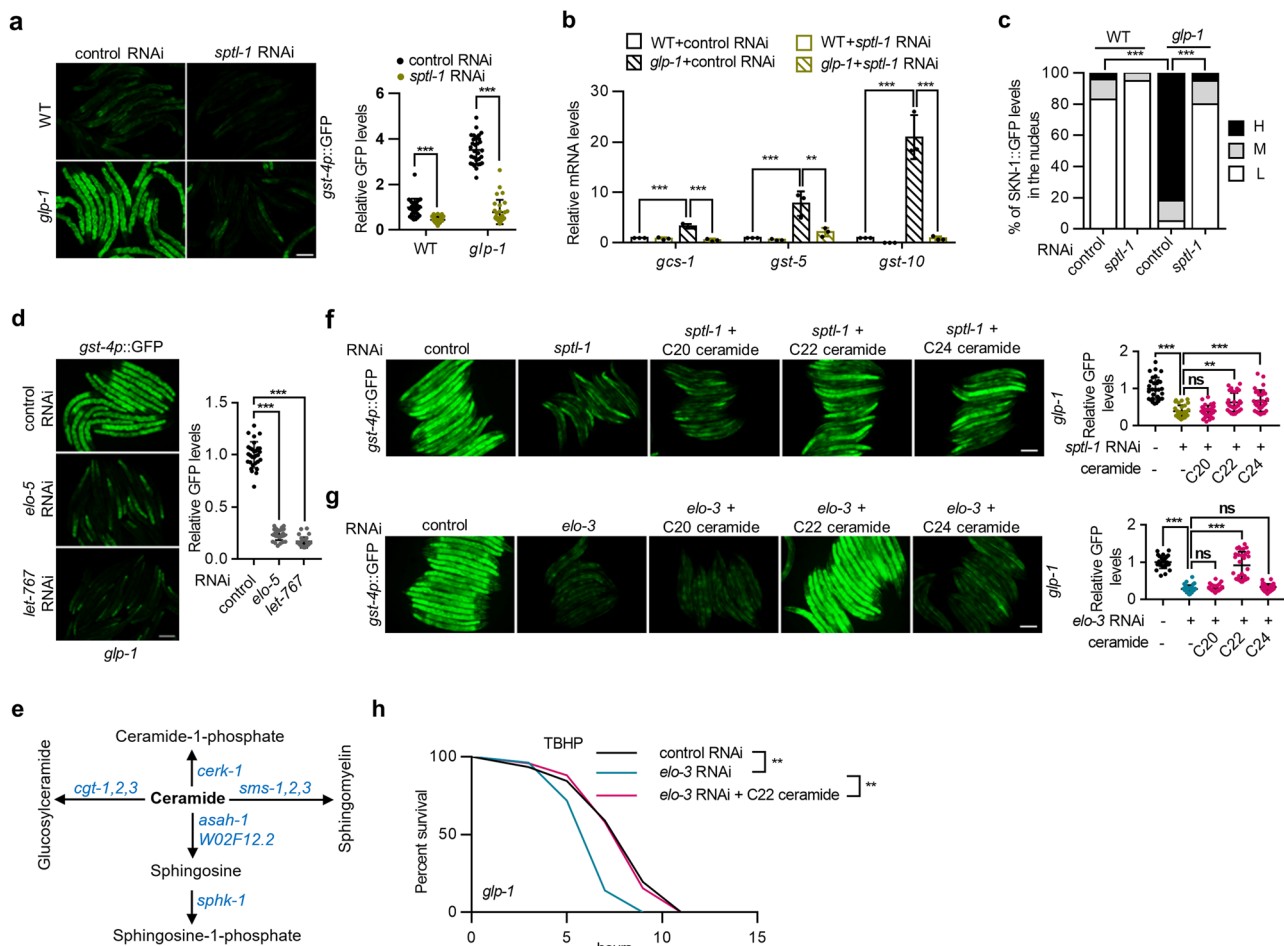

**Fig. 3 C22 ceramide regulates SKN-1 activation and longevity in response to germline loss. a–c** Effects of *sptl-1* RNAi treatment on the expression of *gst-4p*::GFP (**a**) (*n* = 30 animals) and SKN-1 target genes (**b**) (*n* = 3 independent experiments), and SKN-1::GFP nuclear accumulation (**c**) [*n* = 112/89/74/81 animals for WT(control RNAi)/WT(*sptl-1* RNAi)/*glp-1*(control RNAi)/*glp-1*(*sptl-1* RNAi)]. **d** Effects of *elo-5* and *let-767* RNAi treatment on the expression of *gst-4p*::GFP in *glp-1* mutants. *n* = 30 animals. **e** Schematic of ceramide metabolism. **f, g** Effects of ceramide supplementation on *gst-4p*::GFP expression in *glp-1* mutants treated with *sptl-1* (**f**) or *elo-3* (**g**) RNAi. n = 30 animals. **h** Effects of C22 ceramide supplementation on TBHP resistance in *glp-1* mutants treated with *elo-3* RNAi. Data are represented as mean ± SD. **\*\*p < 0.01, \*\*\*p < 0.001. a** was analyzed by two-way ANOVA with Turkey's multiple-comparison test (\*\*\*p < 0.001). **b** was analyzed by one-way ANOVA with Turkey's multiple-comparison test (\*\*\*p < 0.001, \*\*p = 0.0031). **c** was analyzed by Chi-square and Fisher's exact test (\*\*\*p < 0.001). **d, f**, and **g** were analyzed by one-way ANOVA with Turkey's multiple-comparison test (\*\*\*p < 0.001, \*\*p = 0.0013). See Supplementary Table 1 for statistical analysis and additional repeats of the survival assays (**h**) that were analyzed by log-rank (Mantel–Cox) test. Scale bar = 100 μm for panels **a, d, f**, and **g**. Source data are provided as a Source Data file.

metabolism to sphingolipid classes (Fig. 3e), we reasoned that inhibition of ceramide metabolism can theoretically increase ceramide content and thus rescue the phenotypes abrogated by *sptl-1* RNAi. To test this hypothesis, we performed double-RNAi knockdown of *sptl-1* and each ceramide metabolic enzyme simultaneously. As expected, knockdown of ceramide metabolic genes, including *asah-1, cerk-1, sphk-1*, and *W02F12.2*, reversed the suppressive effects of *sptl-1* RNAi on *gst-4p*::GFP expression (Supplementary Fig. 3f). Second, we supplemented worms with commercially available ceramides containing different lengths of acyl chains and found that C22 ceramide (the ceramide containing BA, d18:1/22:0) and C24 ceramide (d18:1/24:0) restored the activity of *gst-4p*::GFP in *sptl-1* RNAi-treated worms (Fig. 3f). As a control, the addition of BA failed to rescue the effects of *sptl-1* RNAi treatment (Supplementary Fig. 3g). Together, these data strongly support the hypothesis that SPTL-1 regulates SKN-1 activation via ceramide production in germline-deficient animals.

These results support the hypothesis that ELO-3-generated BA is the source of the acyl chain in ceramide. To confirm this finding, we performed double-RNAi experiments to knock down

*elo-3* and each ceramide metabolic enzyme. RNAi-targeting ceramide metabolic genes greatly restored the induction of *gst-4p*::GFP in *elo-3* RNAi-treated worms (Supplementary Fig. 3h). Moreover, supplementation with C22 ceramide, but not C20 or C24 ceramide, completely reversed the inhibition of *gst-4p*::GFP expression (Fig. 3g) and oxidative stress resistance (Fig. 3h) in *elo-3*-knockdown animals. We further examined the sufficiency of C22 ceramide in this effect, and the results showed that dietary C22 ceramide induced the expression of *gst-4p*::GFP and SKN-1 targets, enhanced stress resistance, and extended lifespan in germline-intact wild-type animals (Supplementary Fig. 3i–l). In addition, the improvement of stress resistance was dependent on SKN-1, as *skn-1* RNAi completely abrogated the effects of C22 ceramide on TBHP resistance (Supplementary Fig. 3k). Therefore, we conclude that ELO-3 regulates SKN-1 activity and longevity through the generation of BA, which is the source of the acyl chain in C22 ceramide. Since ceramide species in *C. elegans* share the same sphingoid base[28,29] and differ only in the acyl chain, ELO-3 is especially important for the generation of aging-modulating C22 ceramide.

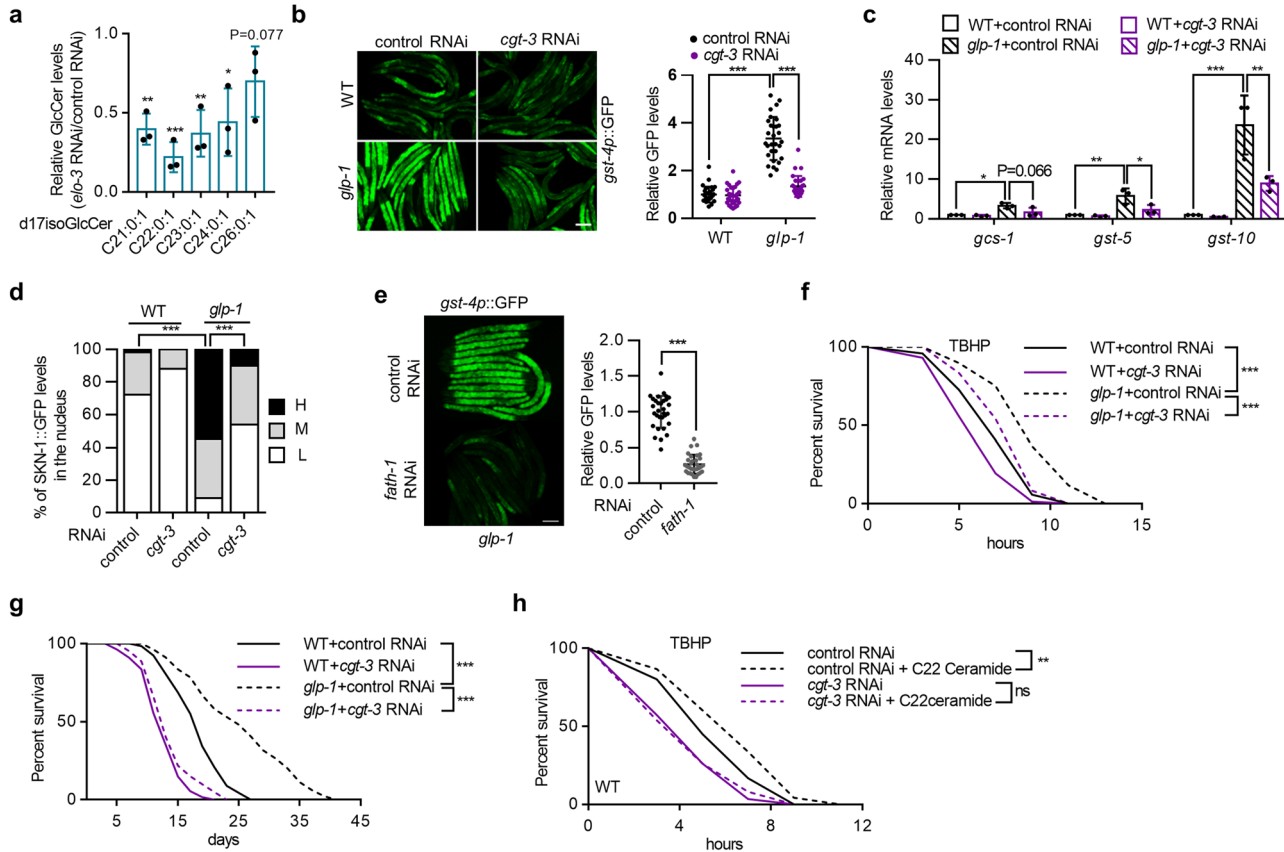

**Fig. 4 C22 GlcCer regulates SKN-1 activity and longevity in response to germline loss. a** Effects of *elo-3* RNAi treatment on the contents of major GlcCer species. $n = 3$ independent experiments. **b–d** Effects of *cgt-3* RNAi treatment on the expression of *gst-4p*::GFP (**b**) ($n = 30$ animals) and SKN-1 targets (**c**) ($n = 3$ independent experiments), as well as SKN-1 nuclear occupancy (**d**) [$n = 99/111/79/92$ animals for WT(control RNAi)/WT(*cgt-3* RNAi)/*glp-1*(control RNAi)/*glp-1* (*cgt-3* RNAi)] in wild-type and *glp-1* mutant worms. (**e**) *fath-1* RNAi treatment suppresses the expression of *gst-4p*::GFP in *glp-1* mutants. $n = 32$ animals. **f, g** Effects of *cgt-3* RNAi on TBHP resistance (**f**) and longevity (**g**) in wild-type and *glp-1* mutant worms. **h** Effects of C22 ceramide addition on TBHP resistance in control and *cgt-3* RNAi-treated wild-type worms. Data are represented as mean ± SD. *$p < 0.05$, **$p < 0.01$, ***$p < 0.001$. **a** was analyzed by multiple *t*-test with correction for multiple comparisons using the Holm–Sidak method (***$p < 0.001$, **$p = 0.0018/0.0053$ for C21:0:1/ C23:0:1, *$p = 0.0206$). **b** was analyzed by two-way ANOVA with Turkey's multiple-comparison test (***$p < 0.001$). **c** was analyzed by one-way ANOVA with Turkey's multiple-comparison test (***$p < 0.001$, **$p = 0.0040/0.0069$ for *gcs-5/gcs-10*, *$p = 0.0137/0.0225$ for *gcs-1/gcs-5*). **d** was analyzed by Chi-square and Fisher's exact test (***$p < 0.001$). **e** was analyzed by unpaired two-tailed *t*-test (***$p < 0.001$). See Supplementary Table 1 and 2 for statistical analysis and additional repeats of the survival assays (**f, g, h**) that were analyzed by log-rank (Mantel–Cox) test. Scale bar = 100 μm for panels **b** and **e**. Source data are provided as a Source Data file.

**C22 glucosylceramide regulates SKN-1 in germline-deficient animals**. Ceramides can function either as signaling molecules or as backbones of other complex sphingolipids (Fig. 3e). Therefore, we asked whether specific sphingolipids derived from C22 ceramide might account for SKN-1 activation and longevity. We used RNAi to knock down ceramide metabolic genes, including *asah-1*, *cerk-1*, *sms-1/3*, *sphk-1*, and *W02F12.2*, and the results showed that none of these genes had any effect on *gst-4p*::GFP expression (Supplementary Fig. 4a), thus ruling out their corresponding sphingolipid classes as activators of SKN-1. The only metabolic pathway left unexplored was the ceramide glycosylation pathway that generates glucosylceramides (GlcCer), which contain acyl chains mainly composed of SVLCFAs[29] and regulate the development of *C. elegans*[32]. Indeed, *elo-3* knockdown dramatically decreased the contents of several major GlcCer species, including BA-containing GlcCer (C22 GlcCer) (Fig. 4a). In addition, the levels of other sphingolipids, such as ceramides and sphingomyelins with side-chain SVLCFAs, were also severely decreased (Supplementary Fig. 4b, c). Surprisingly, the levels of less-abundant C18 ceramide and C18 sphingomyelin were increased upon *elo-3* RNAi treatment (Supplementary Fig. 4b, c),

which may compensate for the decrease in SVLCFA-containing sphingolipids.

We next tested the involvement of ceramide glycosylation in SKN-1 activation. The *C. elegans* genome contains three ceramide glycosyltransferase (CGT) genes, *cgt-1/2/3* (Fig. 3e), and we found that *cgt-3* RNAi (Fig. 4b), but not *cgt-1* or *cgt-2* RNAi (Supplementary Fig. 4d, e), greatly suppressed *gst-4p*::GFP expression in *glp-1* mutants, indicating that CGT-3 was critical for SKN-1 activation. Next, we utilized double-RNAi treatment to detect possible functional redundancy among *cgt* genes. Treatment with RNAi targeting *cgt-3* together with *cgt-1* or *cgt-2* led to *gst-4p*::GFP inhibition, whereas treatment with double *cgt-1* and *cgt-2* RNAis had no effect (Supplementary Fig. 4f). These results are consistent with the major role of CGT-3 in the synthesis of GlcCer in *C. elegans*[33]. We confirmed the involvement of GlcCer by examining the expression of SKN-1 target genes (Fig. 4c) and the nuclear occupancy of SKN-1::GFP (Fig. 4d) upon *cgt-3* knockdown. In *C. elegans*, the fatty acid 2-hydrolase FATH-1 generates 2-hydroxy fatty acids that are incorporated into GlcCer as acyl chains[29], and *fath-1* RNAi similarly reduced *gst-4p*::GFP activity (Fig. 4e), further supporting the idea that GlcCer is

required for SKN-1 induction. The longevity of germline-less animals also required CGT-3, as the knockdown of *cgt-3* compromised the enhanced stress resistance and extended lifespan of *glp-1* mutants (Fig. 4f, g). Moreover, dietary C22 ceramide-induced enhancement of oxidative stress resistance also depended on *cgt-3* (Fig. 4h), suggesting that CGT-3 acts downstream of C22 ceramide, likely via the generation of C22 GlcCer. Therefore, C22 GlcCer is likely the causative sphingolipid in SKN-1 activation and lifespan extension in response to germline absence.

**C22 GlcCer regulates SKN-1 via clathrin.** Glycosphingolipids (GSLs) are the main members of membrane lipids, which affect cellular signal transduction through interactions with membrane-associated proteins. We therefore performed a candidate RNAi screening approach targeting potential membrane proteins. We selected the worm orthologs of mammalian membrane-associated proteins[34]. A total of 508 candidates were tested, and eight genes were found to be required for *gst-4p*::GFP activation in germline-less *C. elegans* (Supplementary Fig. 5a).

Most of the identified gene candidates are nonspecific regulators (Supplementary Fig. 5a). The *chc-1* gene, encoding the heavy chain of the clathrin protein, caught our attention. Clathrin is associated with specific membrane microdomains, plays a major role in the formation of coated vesicles, and regulates multiple cellular processes, such as endocytosis and vesicle trafficking[35]. However, its physiological role in the aging process remains unknown. Since *chc-1* RNAi caused developmental arrest, we treated worms with diluted RNAi or initiated RNAi treatment in worms at the L3 larval stage. Both treatments ensured normal development and inhibited *gst-4p*::GFP expression in *C. elegans* with germline loss (Fig. 5a, Supplementary Fig. 5b). *chc-1* knockdown also abolished SKN-1 nuclear occupancy (Fig. 5b) and lifespan extension (Fig. 5c) in the germline-deficient animals, phenocopying the reduction in C22 GlcCer. Notably, *chc-1* RNAi dramatically shortened the lifespan of wild-type animals; therefore, it cannot be ruled out that the effects of *chc-1* RNAi on lifespan may be due to general sickness.

Clathrin is mainly localized on the plasma membrane, as well as organellar membranes, including the trans-Golgi network, endosomes, and lysosomes. Consistently, CHC-1 expression was mainly observed on the apical membrane of *C. elegans* intestine cells, as revealed by the CHC-1::GFP signals[36] (Fig. 5d). Importantly, the perturbations of C22 GlcCer biosynthesis via the knockdown of *elo-3*, *sptl-1*, or *cgt-3* disrupted the membrane expression of CHC-1::GFP, which was restored upon C22 ceramide supplementation (Fig. 5d and Supplementary Fig. 5c), supporting that C22 GlcCer is required for clathrin membrane localization, which is critical for SKN-1 activation.

**C22 GlcCer interacts with cholesterol to regulate clathrin and SKN-1 activity.** GSLs interact with cholesterol to form membrane microdomains, which are platforms for membrane signal transduction. We therefore asked whether C22 GlcCer functions with cholesterol to regulate clathrin. Within lipid rafts, cholesterol acts as glue that holds raft molecules together. Mammalian studies have reported that excessive cholesterol can restore membrane order and microdomain-related signaling in cells with defective lipid rafts[34,37]. Similarly, we found that dietary supplementation with high cholesterol (HC) largely restored CHC-1::GFP localization (Fig. 5e and Supplementary Fig. 5d), *gst-4p*::GFP induction (Fig. 5f and Supplementary Fig. 5e), SKN-1 nuclear occupancy (Fig. 5g and Supplementary Fig. 5f), and lifespan extension (Fig. 5h and Supplementary Fig. 5g), which had been abrogated upon C22 GlcCer reduction. However, HC failed to reverse the

suppression of *chc-1* RNAi on *gst-4p*::GFP expression (Supplementary Fig. 5h). These genetic evidences are consistent with a model showing that C22 GlcCer interacts with cholesterol, likely in the form of lipid rafts, to regulate clathrin localization and downstream SKN-1 activation in germline-deficient *C. elegans*.

**C22 GlcCer regulates lysosome homeostasis.** How does C22 GlcCer and clathrin regulate lifespan? The binding of clathrin to membrane lipids requires clathrin-associated adapter protein (AP) complexes, which determine the functional specificity of clathrin in different subcellular compartments. To provide clues to clathrin-mediated lifespan regulation, we sought to determine which AP regulates the expression of the SKN-1 reporter. By knocking down each of the *C. elegans* AP-encoding genes, we found that the AP-1 component genes *aps-1* and *apm-1* were required for *gst-4p*::GFP expression in *glp-1* mutants (Fig. 6a). In addition, RNAi knockdown of AP-1 genes also disrupted the membrane localization of CHC-1::GFP, as expected (Supplementary Fig. 6a). We next sought to identify AP-1-dependent cellular processes that are required for SKN-1 activation. In *C. elegans*, AP-1 is involved in the regulation of cell polarity[36,38] and the morphology of lysosome-related organelles (LROs)[39]. We first tested the effect of *par-6* RNAi treatment, which causes defects in cell polarity[40], and found that the expression of *gst-4p*::GFP was not affected (Supplementary Fig. 6b), suggesting that cell polarity is not required for SKN-1 activation.

In *C. elegans*, knockdown of the AP-1-encoding gene results in enlarged LROs[39], which are specialized lysosomes that share many features with classic lysosomes. Moreover, clathrin has been found to regulate lysosome homeostasis during prolonged starvation[41]. As lysosomes are hubs of cellular signal transduction, we next examined whether C22 GlcCer and clathrin function through lysosomes.

We first examined the acidity of lysosomes, which is crucial for lysosomal hydrolase activation and lysosome function. By using carboxy-2′,7′-dichlorofluorescein diacetate (cDCFDA), which can be hydrolyzed into fluorescent cDCF under acidic conditions, we found that perturbations to C22 GlcCer biosynthesis via *elo-3/sptl-1/cgt-3* RNAi treatment led to modest increases in the fluorescence signals in germline-deficient animals, indicative of higher lysosome activity (Fig. 6b). Notably, the size of cDCF-positive granules was also found to be larger in worms with reduced C22 GlcCer than in control animals (Fig. 6b), implying enlarged lysosomes. We employed the lysosome reporter NUC-1::mCHERRY[42] and confirmed the enlargement of lysosomes in *elo-3*-RNAi-treated animals (Supplementary Fig. 6c). Similarly, RNAi of *fath-1* (Supplementary Fig. 6d), *chc-1* (Fig. 6c), and AP-1 component genes (Supplementary Fig. 6e) also increased the acidity and size of lysosomes. Cholesterol supplementation, which rescued CHC-1::GFP localization (Fig. 5e and Supplementary Fig. 5d), also reversed the lysosome phenotypes that had been acquired by animals with reduced C22 GlcCer (Supplementary Fig. 6f). Next, we examined the lysosomal localization of clathrin and found that a small portion of lysosomes stained by LysoTracker were also positive for CHC-1::GFP on the membranes, although the signals were much weaker than those on the plasma membrane (Supplementary Fig. 6g). Collectively, these data suggest that C22 GlcCer and clathrin regulate lysosome activity and morphology.

Lysosomes can fuse with autophagosomes (APs) to form large autolysosomes (ALs), in which the sequestered contents are degraded by hydrolases. Lysosomes can be subsequently regenerated from ALs, a process called autophagic lysosome reformation (ALR). Defects in ALR cause enlarged ALs[43]. A mammalian study revealed a crucial role for clathrin in ALR[41]. Therefore, we

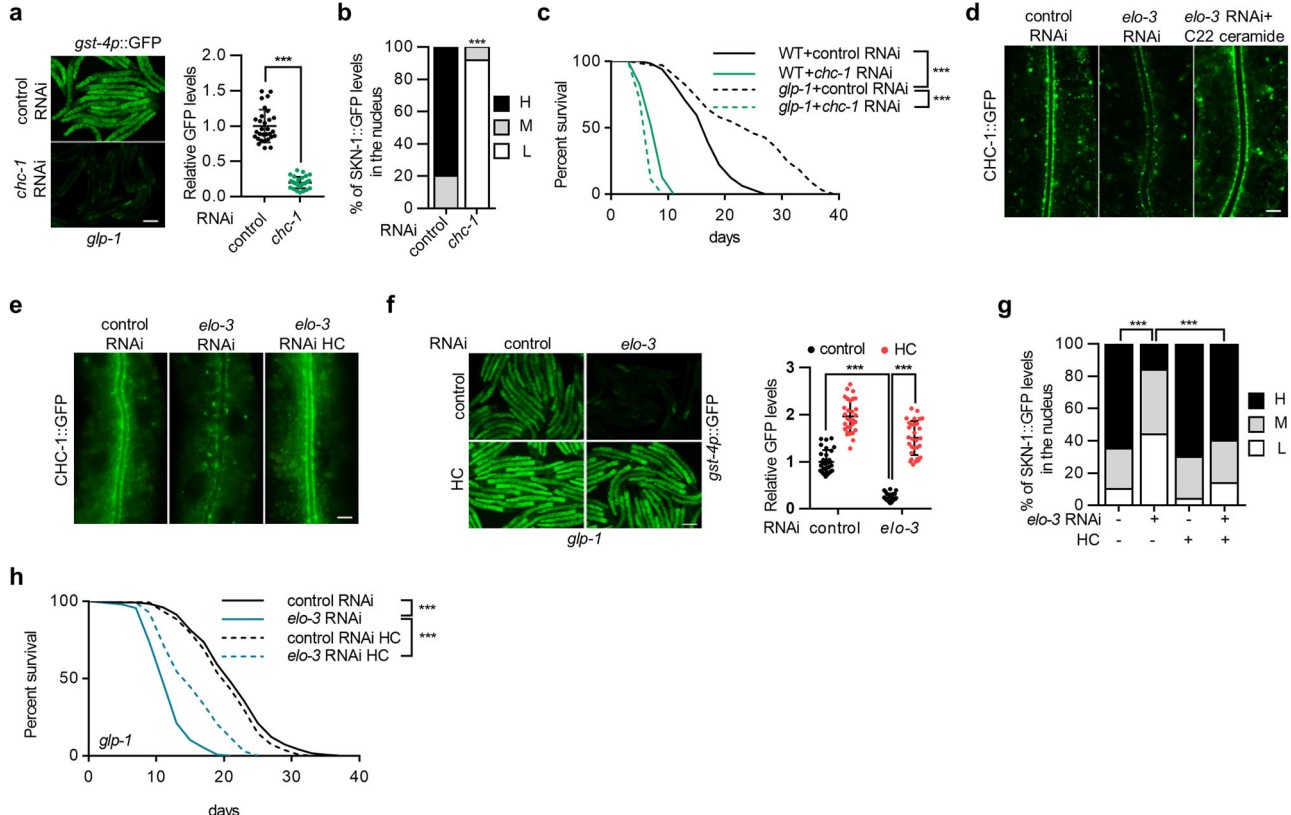

**Fig. 5 C22 GlcCer regulates SKN-1 and longevity via clathrin. a–c** Effects of *chc-1* RNAi treatment of worms from the L3 larval stage on the *gst-4p*::GFP expression (**a**) (*n* = 30 animals), SKN-1::GFP nuclear occupancy (**b**) (*n* = 102/88 animals for control RNAi/*chc-1* RNAi), and longevity (**c**) in *glp-1* mutants. **d** Effects of *elo-3* RNAi and C22 ceramide supplementation on the membrane localization of CHC-1::GFP in the intestine. **e–h** Dietary high cholesterol (HC) rescues the effects of *elo-3* RNAi treatment on CHC-1::GFP membrane localization (**e**), *gst-4p*::GFP expression (**f**) (*n* = 30 animals), SKN-1 nuclear accumulation (**g**) (*n* = 79/80/99/77 animals for control RNAi/*elo-3* RNAi/control RNAi HC/*elo-3* RNAi HC), and lifespan (**h**). Data are represented as mean ± SD. ***p < 0.001. **a** was analyzed by unpaired two-tailed *t*-test (***p < 0.001). **b** and **g** were analyzed by Chi-square and Fisher's exact test (***p < 0.001). **f** was analyzed by two-way ANOVA with Turkey's multiple-comparison test (***p < 0.001). See Supplementary Table 2 for statistical analysis and additional repeats of the survival assays (**c**, **h**) that were analyzed by log-rank (Mantel–Cox) test. Scale bar = 100 μm for panels **a**, **f** and 12 μm for panels **d** and **e**. Source data are provided as a Source Data file.

speculated that the enlarged lysosomes in animals with reduced C22 GlcCer levels might be ALs. Multiple evidence supported the hypothesis. First, enlarged NUC-1::mCHERRY lysosomes were coexpressed with GFP::LGG-1 and GFP::LGG-2 (Fig. 6d and Supplementary Fig. 6h), two marker proteins for APs[44]. To corroborate this finding, we assessed the expression of a mCHERRY::GFP::LGG-1 reporter that indicates ALs with mCHERRY and APs with both GFP and mCHERRY, since GFP fluorescence is quenched in acidic lysosomes[45]. The results showed that a reduction in C22 GlcCer indeed increased the size of the mCHERRY ALs (Fig. 6e). Notably, the fused GFP proteins may respond distinctly in the acidic lysosome environment, as the GFP in mCHERRY::GFP::LGG-1 was quenched in enlarged lysosomes, while the GFP in GFP::LGG-1 and GFP::LGG-2 was not affected. Second, we performed electron microscopy analysis and confirmed the accumulation of enlarged ALs in animals with reduced C22 GlcCer levels (Fig. 6f). Third, RNAis against *bec-1* and *epg-5*, two essential autophagy genes[44,46], disrupted *elo-3* RNAi-induced lysosome enlargement in animals with germline loss (Supplementary Fig. 6i). Together, these findings support the hypothesis that C22 GlcCer reduction leads to the accumulation of enlarged ALs.

We also examined the ALR process directly and observed frequent lysosome reformation from ALs in control animals,

which was rarely seen in the worms with a reduced C22 GlcCer level (Fig. 6g). Moreover, the lysosomes of control animals exhibited massive tubules (likely reflecting active ALR) upon starvation, a condition known to promote ALR, which was also inhibited by a reduction in C22 GlcCer (Fig. 6h). Collectively, these data suggest that reduction in C22 GlcCer impairs ALR, which leads to enlarged ALs.

Enlarged ALs may also result from defects in autophagic degradation. Cathepsins constitute a class of lysosomal enzymes whose activities can be measured by Magic Red staining, which releases red fluorescence upon cathepsin cleavage. Consistent with the cDCF-intensity data, the activity levels of cathepsins were increased in worms with reduced C22 GlcCer production (Supplementary Fig. 6j). We also measured lysosomal activity by examining the lysosomal cleavage of NUC-1::mCHERRY, which was not affected by C22 GlcCer reduction (Supplementary Fig. 6k). These data suggest that the general activity of lysosomes is not impaired by C22 GlcCer reduction. We further examined autophagic degradation activity by measuring LGG-1::GFP processing and found that GFP cleavage in ALs was dramatically suppressed by C22 GlcCer reduction (Supplementary Fig. 6l). Therefore, the ALs in animals with reduced C22 GlcCer levels have impaired degradation activity, which may also contribute to AL enlargement.

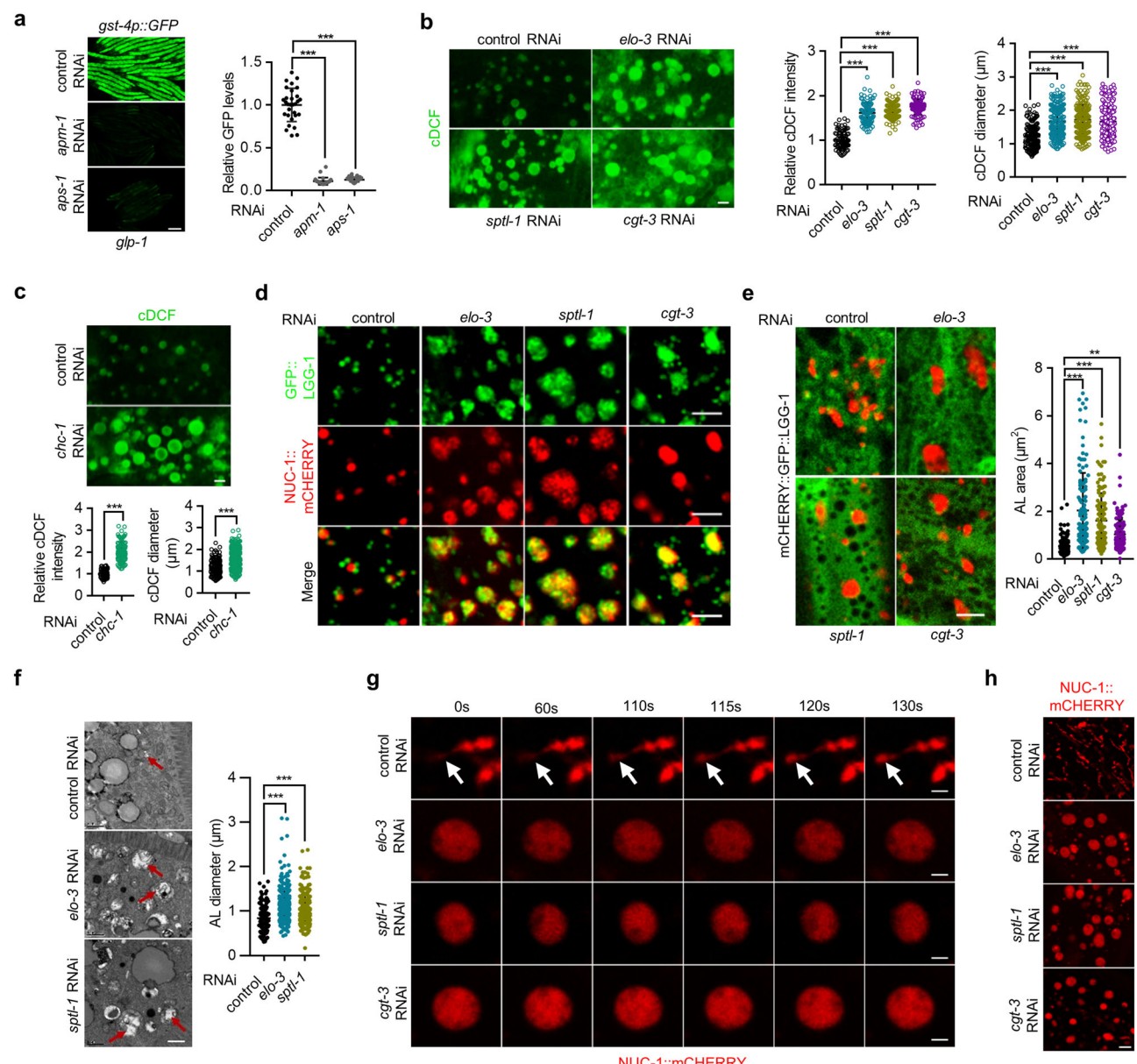

**Fig. 6 C22 GlcCer regulates lysosome homeostasis. a** Effects of *aps-1* and *apm-1* RNAi treatment on the expression of *gst-4p*::GFP in *glp-1* mutants. $n = 30$ animals. **b** Effects of *elo-3*, *sptl-1*, and *cgt-3* RNAi treatment on the acidity and size of the lysosomes in the intestine of *glp-1* mutants, as determined by cDCFDA staining. $n = 105$ (control RNAi, *elo-3* RNAi, and *cgt-3* RNAi) and 104(*sptl-1* RNAi) cDCF-positive granules for intensity analysis; $n = 205$ (control RNAi), 210 (*elo-3* RNAi), 207 (*sptl-1* RNAi), and 211 (*cgt-3* RNAi) cDCF-positive granules for size analysis. **c** Effects of *chc-1* RNAi treatment on the acidity and size of the lysosomes in the intestine of *glp-1* mutants. $n = 108$ (control RNAi) and 113 (*chc-1* RNAi) cDCF-positive granules for intensity analysis; $n = 205$ (control RNAi) and 204 (*chc-1* RNAi) cDCF-positive granules for size analysis. **d** Colocalization between GFP::LGG-1 and NUC-1::mCHERRY in the hypodermis of *elo-3*, *sptl-1* and *cgt-3* RNAi-treated worms. **e** Effects of *elo-3*, *sptl-1*, and *cgt-3* RNAi on the AL area, as measured by the red-only signals from mCHERRY::GFP::LGG-1 in the hypodermis. $n = 125$ ALs for control RNAi and 129 ALs for *elo-3*, *sptl-1*, and *cgt-3* RNAi. **f** Effects of *elo-3* and *sptl-1* RNAi treatment on AL size as revealed by EM analysis of the intestine. Red arrows indicate ALs. $n = 115/289/232$ ALs for control RNAi/*elo-3* RNAi/*sptl-1* RNAi. **g** Time-lapse images showing the effects of *elo-3*, *sptl-1*, and *cgt-3* RNAi treatment on lysosome reformation from AL in the hypodermis. White arrows indicate newly formed lysosomes. **h** Effects of *elo-3*, *sptl-1*, and *cgt-3* RNAi treatment on lysosome morphology in the hypodermis in response to 2-h starvation. Data are represented as mean ± SD. **$p < 0.01$, ***$p < 0.001$. **a, b, e**, and **f** were analyzed by one-way ANOVA with Turkey's multiple-comparison test (***$p < 0.001$, **$p = 0.0011$ for panel e). **c** was analyzed by unpaired two-tailed *t*-test (***$p < 0.001$). Scale bar = 100 μm for panel **a**, 3 μm for panels **b**, **c**, and **d**, 2.5 μm for panel **e**, 1 μm for panels **f** and **g**, and 4 μm for panel **h**. Source data are provided as a Source Data file.

**C22 GlcCer regulates SKN-1 via TOR.** The physiological roles of lysosome homeostasis are only beginning to emerge, and its involvement in the aging process is unknown. After prolonged fasting, mammalian mTOR is activated, which promotes ALR and terminates autophagy[43], thereby maintaining lysosome homeostasis. We speculated that impaired ALR and/or accumulation of ALs resulting from reduced C22 GlcCer levels may signal the cell to promote ALR, in order to restore lysosome homeostasis. Therefore, C22 GlcCer reduction might induce TOR activation, which is known to negatively regulate the lifespan in *C. elegans* with germline loss[47]. Moreover, the suppression of SKN-1 by the TOR pathway in *C. elegans* has been established. TORC1

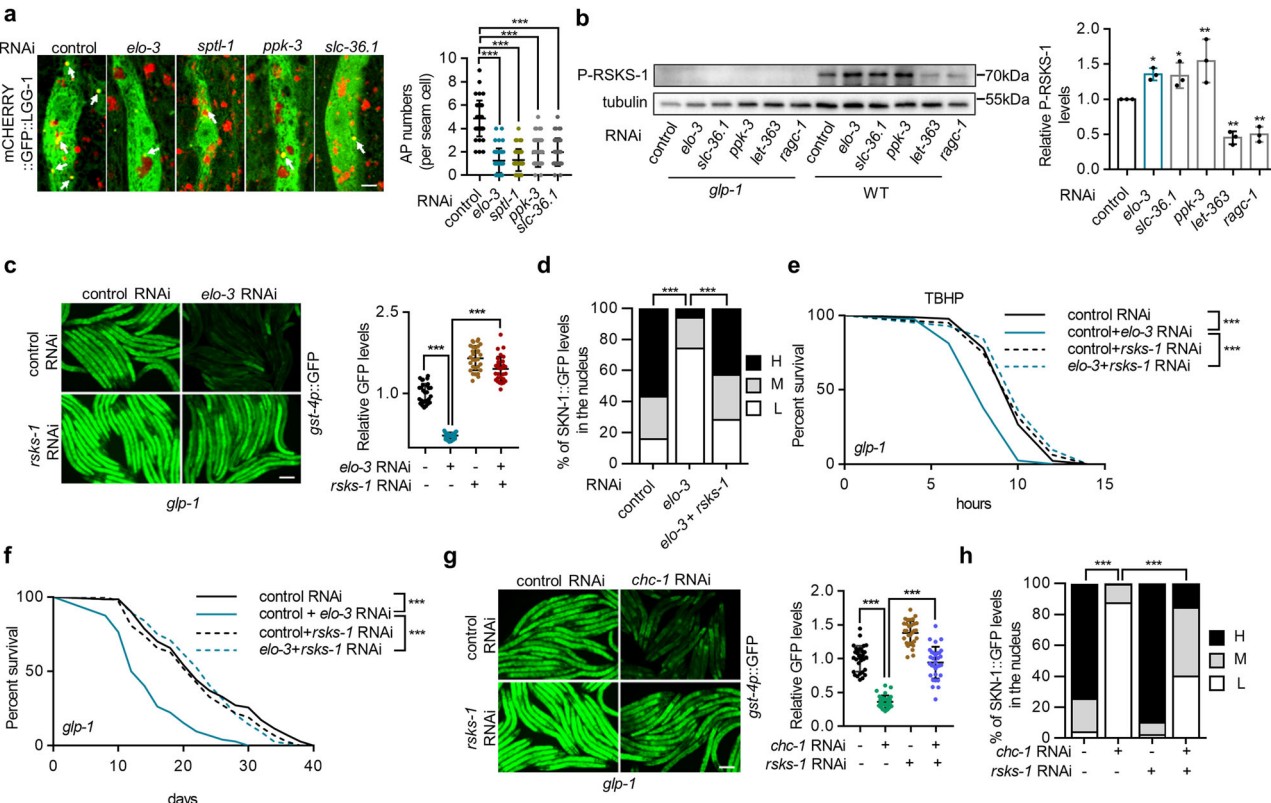

**Fig. 7 C22 GlcCer regulates SKN-1 and longevity via TOR. a** Effects of *elo-3*, *sptl-1*, *slc-36.1*, and *ppk-3* RNAi treatment on the number of APs in seam cells as measured by yellow signals (red + green) from mCHERRY::GFP::LGG-1. White arrows indicate APs. n = 42/44/45/42/42 seam cells for control RNAi/ *elo-3* RNAi/*sptl-1* RNAi/*ppk-3* RNAi/*slc-36.1* RNAi. **b** Effects of RNAi of *elo-3*, *slc-36.1*, and *ppk-3* and TOR component genes on the phosphorylation of RSKS-1 as measured by immunoblot analysis. n = 3 independent experiments. **c–f** *rsks-1* RNAi treatment restores the expression of *gst-4p*::GFP (**c**) (n = 30 animals), SKN-1 nuclear occupancy (**d**) (n = 72/91/89 animals for control RNAi/*elo-3* RNAi/*elo-3* + *rsks-1* RNAi), TBHP resistance (**e**), and lifespan (**f**) in *glp-1* mutants treated with *elo-3* RNAi. **g, h** *rsks-1* RNAi treatment restores the expression of *gst-4p*::GFP (**g**) (n = 32 animals) and SKN-1 nuclear occupancy (**h**) (n = 82/79/99/89 animals for control RNAi/*chc-1* RNAi/*rsks-1* RNAi/*chc-1* + *rsks-1* RNAi) in *glp-1* mutants with *chc-1* knockdown. Data are represented as mean ± SD. **p < 0.01, ***p < 0.001. **a**, **c**, and **g** were analyzed by one-way ANOVA with Turkey's multiple-comparison test (***p < 0.001). **b** was analyzed by one-way ANOVA with Holm–Sidak's multiple-comparison test (**p = 0.0065/0.0065/0.0074 for *ppk-3*/*let-363*/*ragc-1* RNAi, *p = 0.0376/ 0.0376 for *elo-3*/*slc-36.1* RNAi). **d** and **h** were analyzed by Chi-square and Fisher's exact test (***p < 0.001). See Supplementary Table 1 and 2 for statistical analysis and additional repeats of the survival assays (**e**, **f**) that were analyzed by log-rank (Mantel-Cox) test. Scale bar = 3 μm for panel **a**, 100 μm for panels **c** and **g**. Source data are provided as a Source Data file.

inhibits SKN-1, at least in part by increasing general mRNA translation, and TORC2 regulates SKN-1 likely via AKT and SGK-1 kinases. RNAi knockdown of TORC1 or TORC2 components extends lifespan in a SKN-1-dependent manner[48]. Thus, we proposed that in worms with reduced C22 GlcCer levels, enlarged ALs might result in TOR activation, which can impair SKN-1 activation and reduce longevity.

To test this hypothesis, we first examined AP formation, which is negatively regulated by TOR. We again assessed the mCHERRY::GFP::LGG-1 reporter, which was used to distinguish between APs and ALs. ALs were considered inappropriate for the evaluation of TOR activity in the current study, because TOR was proposed to be activated by the accumulation of ALs. Impaired C22 GlcCer biosynthesis indeed reduced the number of APs (Fig. 7a). Electron microscopy analysis confirmed this AP reduction (Supplementary Fig. 7a). We also verified the mCHERRY::GFP::LGG-1 result by assessing the effect of the *epg-5* mutation, which blocks the fusion of APs and lysosomes[46]. As expected, *epg-5* mutation increased the AP numbers in both control and *elo-3/sptl-1* RNAi-treated animals (Supplementary Fig. 7b) and partially suppressed the formation of enlarged ALs in animals with reduced C22 GlcCer levels (Supplementary Fig. 7c). Second, we examined TOR activity by measuring the

phosphorylation of its substrate S6 kinase (S6K). We failed to detect phosphorylated RSKS-1 (the worm ortholog of S6K) in germline-deficient animals through immunoblot analysis, possibly due to its low expression in the *C. elegans* soma. Instead, we observed increased phosphorylation of RSKS-1 in wild-type animals treated with *elo-3* RNAi (Fig. 7b). These data suggest that C22 GlcCer is a negative regulator of the TOR pathway.

To further prove the idea that the accumulation of enlarged ALs can activate TOR, we knocked down ALR genes, including *slc-36.1* and *ppk-3*[49], which leads to enlarged ALs (Supplementary Fig. 7d). As expected, RNAi treatment of these ALR genes induced TOR activation, as indicated by the decrease in AP formation (Fig. 7a) and increase in RSKS-1 phosphorylation (Fig. 7b). Therefore, ALR and ALs likely bridge the C22 GlcCer and TOR pathways.

We further tested whether C22 GlcCer regulates SKN-1 via TOR suppression. RNAi targeting TOR component genes *rsks-1*, *raga-1*, *ragc-1*, *rheb-1*, *let-363*, and *daf-15* largely restored the induction of *gst-4p*::GFP in *elo-3*-knockdown animals (Fig. 7c and Supplementary Fig. 7e). Consistently, treatment with *rsks-1* and *ragc-1* RNAis also reversed the effects of *elo-3* knockdown on SKN-1 nuclear occupancy (Fig. 7d and Supplementary Fig. 7f), oxidative stress resistance (Fig. 7e and Supplementary Fig. 7g),

and worm longevity (Fig. 7f and Supplementary Fig. 7h). These results suggest that TOR suppression mediates the C22 GlcCer effects on SKN-1 activation and longevity. Additionally, *rsks-1* knockdown restored the induction of *gst-4p*::GFP (Fig. 7g and Supplementary Fig. 7i) and SKN-1 nuclear occupancy (Fig. 7h and Supplementary Fig. 7j) in animals treated with RNAis targeting clathrin and AP-1 components. However, neither the lysosome morphology nor the CHC-1::GFP localization in *elo-3*-knockdown animals was reversed by *rsks-1* or *ragc-1* RNAi treatment (Supplementary Fig. 7k, l), consistent with the model showing that TOR signaling acts downstream of clathrin and lysosomes to regulate SKN-1 activation.

## Discussion

The present study defined a sphingolipid-based longevity pathway that is determined by a specific acyl chain. The lifespan-extension effects of germline loss depend on TOR inhibition, which requires normal lysosome homeostasis that is critically regulated by BA, C22 GlcCer, and clathrin. Without a single saturated fatty acid BA or its derived C22 GlcCer, clathrin localization and lysosome homeostasis are disrupted at the step of ALR. This disruption leads to the formation of giant ALs, which activate TOR and reduce longevity (Fig. 8).

We clarified a metabolic cascade for the generation of aging-modulating C22 GlcCer. Specifically, LET-767 and ELO-5 catalyze the generation of mmBCFAs, which are key components in the sphingoid base of ceramide produced by SPTL-1. In parallel, ELO-3-generated BA is incorporated into ceramide as the acyl chain. Then, CGT-3 catalyzes the glycosylation of C22 ceramide to generate C22 GlcCer. The current study suggests that these acyl chains may endow sphingolipids with remarkable and specific regulatory functions in the aging process. In line with this, C20 and C22 ceramides can block the cytosolic heat shock response induced by the suppression of nematode mitochondrial chaperone expression[50], and C24 ceramide is essential for mitochondrial surveillance[51]. Notably, C24 ceramide may also regulate SKN-1 activation, as it restored *gst-4p*::GFP expression in *sptl-1* RNAi-treated animals (Fig. 3f) but not in *elo-3* RNAi-treated animals (Fig. 3g). Since it has been reported that ceramides with different acyl chains can function oppositely[52], a possible explanation is that the increased C18 ceramide (Supplementary Fig. 4b) may counteract the effects of C24 ceramide in *elo-3* RNAi-treated animals.

The current study links GSLs to longevity response. Mammalian studies have reported that the contents of multiple GSLs, including C22 GSLs, are increased in the plasma and organs and particularly in the brains of aged animals[53,54]. In contrast, aging is associated with a reduction in GSL levels in immune cells, which accounts for aging-related impairment of immune cell function[55]. Therefore, GSL species may exert opposite effects on aging-related phenotypes in a tissue-specific manner. This hypothesis is consistent with the pleiotropic roles of TOR, which is negatively regulated by C22 GlcCer, in aging and organ functions. For instance, TOR inhibition has beneficial effects on immune cell functions[56], whereas TOR activity is required for the memory function in the brain[57].

A previous study in *C. elegans* reported that GlcCer appears to be a positive regulator of TORC1 expression during development[32,58], which seems to contradict our findings. We speculated that GlcCer might control TOR regulation in an acyl chain-dependent and/or developmental stage-dependent manner. As TOR is a central regulator of numerous biological processes, including cell growth and metabolism, our study may have general implications for the metabolic regulation of TOR and TOR-related processes.

Clathrin controls the availability of multiple membrane receptors via endocytosis. Accordingly, clathrin-mediated endocytosis has been implicated in many receptor-dependent cellular functions, including cell growth and differentiation. Further studies exploring the roles of BA and C22 GlcCer in these biological processes would be interesting.

The amount of cholesterol in the lipid extracts of worms is much less than that in mammals[59], suggesting that cholesterol may not be a major structural component of the cell membrane in *C. elegans*. However, the genetic interaction between C22 GlcCer and cholesterol supports the existence of GSL-enriched microdomains in *C. elegans*, which likely function in lysosomes to regulate lifespan. This finding is also supported by the finding that cholesterol is mainly located in intracellular compartments, including LROs, in the intestinal cells of *C. elegans*[60].

In summary, the present study reveals that an acyl chain-specific sphingolipid can regulate longevity and suggests a mechanistic link between membrane lipids and aging. Given the conservation of fatty acid and sphingolipid metabolism, these lipid species may also have beneficial effects on longevity of other species, including mammals.

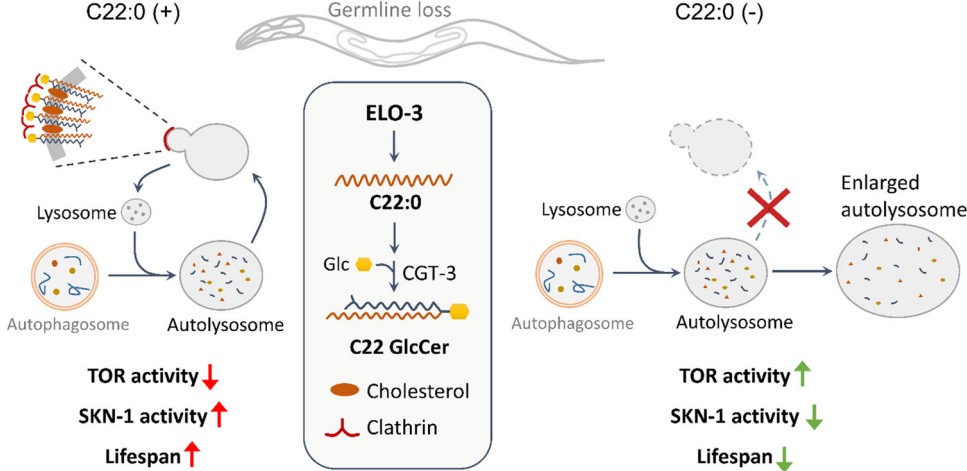

**Fig. 8 Model of lifespan regulation by C22:0.** ELO-3 regulates the production of BA that is incorporated into C22 GlcCer as the acyl chain (middle). In germline-deficient animals, C22 GlcCer regulates clathrin membrane localization and lysosome homeostasis, which is required for TOR inhibition, SKN-1 activation, and lifespan extension (left). In germline-deficient animals with reduced C22 GlcCer production, clathrin localization and lysosome regeneration are disrupted, leading to TOR activation and lifespan reduction (right).

## Methods

**C. elegans strains and maintenance**. *C. elegans* were cultured on standard nematode-growth medium (NGM) seeded with *E. coli* OP50-1[61]. The following strains were provided by Caenorhabditis Genome Center: wild-type N2 Bristol, CB4037[*glp-1(e2141)*], DR1572[*daf-2(e1368)*], DA465[*eat-2(ad465)*], CL2166[*gst-4p::gfp*], LD1[*skn-1b/c::gfp + rol-6(su1006)*], CF1553[*sod-3p::gfp + rol-6(su1006)*], TJ375[*hsp-16.2p::gfp*], DA2123[*lgg-1p::gfp::lgg-1 + rol-6(su1006)*], and MAH215 [*lgg-1p::mCherry::gfp::lgg-1*]. XW5399[*ced-1p::nuc-1::mcherry*] was provided by Dr. Xiaochen Wang, strain containing the array *yqEx480* (*chc-1p::chc-1::gfp*) was provided by Dr. Chonglin Yang and strain containing the array *bpIs220* (*lgg-2p::gfp::lgg-2*) and *epg-5(tm3425)* was provided by Dr. Hong Zhang. *glp-1 (e2141)* mutants are sterile and long-lived when raised at the nonpermissive temperature of 25 °C during larval stage. Double mutants were generated by standard genetic techniques.

**Microbe strains**. *E. coli* OP50-1 bacteria were cultured overnight at 37 °C in LB, after which 150ul of bacterial cultures were seeded on 60-mm NGM plates. For RNAi experiment, HT115 bacteria containing specific dsRNA-expression plasmids (Ahringer library) were cultured overnight at 37 °C in LB containing 100ug/ml carbenicillin, and seeded onto NGM plates containing 5 mM IPTG[62].

**RNA-interference treatment**. RNAi was induced at room temperature for 24 h after seeding. Then, L1 worms were added to RNAi plates to knock down indicated genes. For *elo-5*, *let-767*, and *chc-1* RNAi treatments, the bacterial cultures of target RNAi strains were diluted with the vector control at the ratios of 1:5, 1:15, and 1:50, respectively. For double RNAi, two RNAi strains were mixed at a ratio of 1:1.

**qRT-PCR**. qRT-PCR was performed as previously described[63]. Briefly, day-1 adult worms were collected, washed in M9 buffer, and then homogenized in Trizol reagent (Life Technologies). RNAs were extracted according to the manufacturer's protocol. DNA contamination was digested with DNase I (Thermo Fisher Scientific) and RNA was subsequently reverse-transcribed to cDNA by using the RevertAid First Strand cDNA synthesis Kit (Thermo Fisher Scientific). Quantitative PCR was performed using SYBR Green (Bio-Rad) and data were collected by CFX Maestro Software. The expression of *cdc-42* was used to normalize samples. Primer sequences were listed in Supplementary Table 3.

**Lifespan analysis**. Lifespan assays were performed as previously described[64]. Briefly, synchronized L1 worms were added to NGM plates seeded with different *E. coli* strains. *glp-1* mutants were kept at 25 °C from L1 to day-1 adult, and shifted to 20 °C thereafter. Worms were transferred every day during the reproductive period. Worms that died of vulva burst, bagging, or crawling off the plates were censored.

**Oxidative stress resistance assays**. For TBHP (Sigma) resistance, day 1 adult worms were transferred to NGM plates supplemented with 12.6 mM TBHP and incubated at 20 °C for survival analysis.

**Fluorescent microscopy**. To analyze GFP or mCHERRY fluorescence, day-1 adult worms were paralyzed with 1 mM levamisole and fluorescent microscopic images were taken after being mounted on slides by Nikon NIS-Elements software or Leica LAS X software. To study the SKN-1 nuclear localization, SKN-1B/C::GFP worms were mounted on slides. The levels of GFP nuclear localization were scored. Briefly, no nuclear GFP, GFP signal in the nucleus of anterior or posterior intestine cells, and nuclear GFP in all intestinal cells are categorized as low, medium, and high expression, respectively. To study NUC-1::mCHERRY dynamics, worms were fasted for 2 h to facilitate ALR and images were captured every 5 sec for 3 min. GFP and cDCFDA intensities were analyzed by Image J 1.53e software. cDCFDA granule size was analyzed by Nikon NIS-Elements software and AL size from mCHERRY::GFP::LGG-1 was analyzed by Leica LAS X software.

**cDCFDA, Lyso–Tracker, and Magic Red staining**. For cDCFDA staining, 100 μL of 1.5 mM cDCFDA solution was added to the surface of bacterial lawn. Worms at day-1 adult stages were transferred to the cDCFDA plates and cultivated for 16 h, after which fluorescent microscopic images were taken in the GFP channel. For Lyso-Tracker red staining, 100 μL of 10 μM Lyso-Tracker red solution was added to the surface of bacterial lawn. Adult worms were transferred to the Lyso-Tracker red plates and cultivated for 1 h before imaging. For magic red staining (ImmunoChemistry Technologies), stock solution was prepared following the manufacturer's instruction. 2 μL of stock was diluted in 48 μL of M9 and was added to the surface of bacterial lawn. Adult worms were added to the magic red plates and cultivated for 12 h before imaging.

**Transmission electron microscopy**. Day-1 adult worms were prefixed in 2.5% glutaraldehyde/2% PFA for 1 h at room temperature and stored at 4 °C, then worms were cut into pieces and postfixed and stained. Briefly, worms were postfixed in 1% OsO$_4$/3% K$_3$[Fe(CN)$_3$] for 1 h at 4 °C, then washed and fixed in 1% thiocarbohydrazide for 30 min, and then washed and fixed again in 1% OsO$_4$ for 30 min. Fixed worms were stained with 2% uranyl acetate reagent overnight at 4 °C. Then worms were preembedded in 2% agarose, dehydrated in ethanol/acetone series, infiltrated and embedded in EMBed-812 resin. Samples were cut into 70-nm sections with microtome EM UC7 (Leica Bio-systems) and observed with JEM-1230 (JEOL) operating at 80 kV.

**Immunoblotting**. Day-1 adult worms were collected and sonicated in RIPA buffer (100 mM Tris, pH 8.0, 150 mM NaCl, 1% Triton X-100, 1% deoxycholic acid, 0.1% SDS, 5 mM EDTA, and 10 mM NaF) containing 1 mM DTT and proteinase inhibitor (Sigma) before boiling and loading. Antibodies against GFP (Santa Cruz, SC-9996, 1:2000), mCherry (Sungene Biotech, KM8017, 1:2000), phospho-p70 S6K (Cell signaling, #9205, 1:1000), and tubulin (Sigma, T9026, 1:4000) were used. The images were quantified by Image J 1.53e.

**Fatty acid quantification**. Fatty acid contents were measured as previously described[65] with some modifications. About 500–1000 age-synchronized day-1 adult worms were washed off plates and washed three times with water. Worm pellets were resuspended with 1.2 mL of 2.5% H$_2$SO$_4$ in methanol and incubated at 80 °C for 1 h. Then, 1 ml of supernatant was mixed with 1.2 ml of hexane and 1.8 ml of water to extract fatty acid methyl esters (FAMEs) for GC–MS/MS analysis. The Supelco 37 Component FAME Mix (Sigma) was used to determine the retention time. The *Shimadzu GCMS-TQ8040* Gas Chromatograph Mass Spectrometer equipped with SH-Rxi-5sil MS column was used. The contents of fatty acids were normalized to protein concentrations. The data were collected by GCMSsolution software.

**Biochemical quantifications of triglyceride**. Lipid quantitation was conducted as previously described[66,67] with some modifications. About 5000 age-synchronized day-1 adult worms were washed with M9 to remove bacteria and then sonicated in 0.25 mL of PBS. Lipids were exacted with 1.5 mL of chloroform: methanol (2:1) through centrifugation. The lower organic phase was recovered and added with 3 mL of 0.9% NaCl to separate the phase. After centrifugation, the lower organic phase was recovered again and evaporated under nitrogen. Dried lipids were resuspended in 1 ml of chloroform for solid-phase exchange (SPE) chromatography. The SPE columns (Fisher Scientific) were preequilibrated with 3 ml of chloroform and loaded with extracted lipid. Triglycerides were first eluted with 3 ml of chloroform, followed by glycosphingolipids eluted with 5-ml mixture of 9:1 acetone:methanol. Phospholipids were eluted with 3 ml of methanol. Purified lipids were evaporated under nitrogen, resuspended in 2.5% H$_2$SO$_4$ in methanol, and incubated for 1 h at 80 °C to create FAMEs for GC–MS/MS analysis.

**Sphingolipid extraction and quantification**. Lipid quantitation was conducted as previously described[32,68,69] with some modifications. Day-1 adults were washed from plates and rinsed by M9 three times to remove bacteria before being centrifuged. For each sample, 500 μl of worm pellet was collected and 100 μg of d18:1/ C4 ceramide (dissolved in methanol, Avanti Polar Lipid) was added as the internal control. The total lipid was extracted as previously described[32]. The crude lipid extract in the organic layer was then dried under a nitrogen stream and 1 M KOH (in methanol) was added to remove triglycerides and glycerophospholipids, as described before. After phase separation, the sphingolipid fraction (the organic phase) was then dried under a nitrogen stream and dissolved in 200 μl (for UHPLC/mass analysis) or 500 μl of solvent A (isopropanol/hexane/100 mM ammonium acetate 58:40:2; v/v) (for Shotgun analysis).

The shotgun analysis was done by SCIEX Qtrap6500 and precursor scan m/ z = +250.3 and m/z = +184.0 (CE = 46) were used to profile all Cer/GlcCer and sphingomyelin respectively. For the MRM-based quantification of sphingolipid molecules, a SHIMADZU LC30A UHPLC was coupled to the SCIEX Qtrap6500. Separation of lipids was achieved on a Phenomenex Luna®3-μm silica column (100-Å LC column, 150*2 mm) using a gradient from solvent A to solvent B (isopropanol/hexane/100 mM ammonium acetate 50/40/10; v/v). Analysis was performed in positive-ionization mode. Gradient conditions were as follow: 50% B from 0 to 5 min. From 5 to 30 min, B was linearly ramped to 100%, where it remained for 10 min. From 40 to 41 min, B was returned to 50%, where it remained until the end of the run at 50 min.

**Lipid supplementation**. Fatty acid supplementation experiments were performed as previously described[70]. Briefly, 22:0 and 21:0 fatty acids were dissolved in DMSO to make 300 mM stock solutions, which were added to NGM before pouring the plates at a final concentration of 600 μM. For ceramide treatment, C20, C22, and C24 ceramides were dissolved in ethanol to make 0.5 mg/ml stock solutions and 60 μg of ceramides were added on the surface of NGM plates before bacteria seeding. For high-cholesterol supplementation, cholesterol was added to NGM at a concentration of 0.1 mg/ml.

**Statistics and reproducibility**. Data are presented as mean ± SD. GraphPad prism 8 software was used for statistical testing. Survival data were analyzed by using a log-rank (Mantel–Cox) test. The nuclear accumulation of SKN-1::GFP was analyzed by using a Chi-square and Fisher's exact test. Other data were analyzed by

using ANOVA or *t*-test as indicated in figure legends. $p < 0.05$ was considered as significant. Micrographic and immunoblotting images are representative of three independent experiments, with similar results.

**Reporting summary**. Further information on research design is available in the Nature Research Reporting Summary linked to this article.

## Data availability

The data supporting the findings from this study are available within the article and its supplementary information. Source data are provided with this paper. Any remaining raw data will be available from the corresponding author upon reasonable request. Source data are provided with this paper.

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

## Acknowledgements

We thank CGC, Hong Zhang, Chonglin Yang, and Xiaochen Wang for providing strains. We thank Qin Deng and Guocan Zheng in Analytic and Testing Center of Chongqing University for use of the facility and technical support for confocal and GC–MS/MS analysis, ShanghaiTech School of Life Science and Technology Core facility, and Ying Han for their assistance with sphingolipid profiling and quantification, and Yu Kong and Lijun Pan (Electron Microscopy Facilities of Center for Excellence in Brain Science and Technology, CAS) for assistance with EM sample preparation and EM image analysis. This work was supported by National Natural Science Foundation of China (grant No. 31771337, 32071163 to S.P. and grant No. 32070754 to H.T.). This work was also supported by Natural Science Foundation of Chongqing, China (cstc2020jcyj-msxmX0714 to H.T.), Chongqing Talents Plan for Young Talents (CQYC201905071 to S.P.), the strategic Priority Research Program of the Chinese Academy of Sciences (XDB39000000 to B.Z.), Youth Thousand Talents Plan (H.Z.), Shanghai Pujiang Program (16PJ1407400), and the National Key R&D Program of China (2019YFA0802804).

## Author contributions

H.T. and S.P. conceived and supervised the study, F.W., Y.D., H.Z., B.Z., H.T., and S.P. designed the experiments, F.W., Y.D., X.Z., and Q.C. performed the experiments, F.W., Y.D., X.Z., Q.C., H.Z., B.Z., H.T., and S.P. analyzed data, and H.T. and S.P. wrote the paper.

## Competing interests

The authors declare no competing interests.
