## [Peer Review File · Nature Communications]

Reviewer comments, first round –

Reviewer #1 (Remarks to the Author):

In this study, the authors identified C22 GlcCer as a key molecule involved in SKN-1 activation and lifespan extension in germline-deficient *C. elegans*. The authors showed that C22 GlcCer is required for membrane localization of clathrin. Defects in C22 GlcCer production causes enlarged autolysosomes. Moreover, TOR activity seems to increase in *elo-3* RNAi worms and inhibiting TOR activity restored SKN-1 reporter gene expression, TBHP resistance and lifespan in *elo-3* RNAi worms. The authors proposed that C22 GlcCer affects membrane protein clathrin and thus modulates lysosome homeostasis, TOR signaling and SKN-1 activation.

This manuscript provided clear data that C22 GlcCer production is important for SKN-1 activation, TBHP resistance and lifespan extension in germline-deficient *glp-1* mutant. However, the mechanistic link between C22 GlcCer, ALR and TOR activity is not established convincingly.

Major comments:

1. For most cellular and molecular assays, data were compared but not quantified.

1) As the key assay for SKN-1 transcriptional activity utilized throughout the whole study, the *gst-4p::GFP* intensity should be quantified and then compared in different strains/conditions to clearly indicate changes in SKN-1 activity.

2) In Fig. 6 and S6, cDCF intensity and lysosome size should be quantified.

3) In Fig 7A and B, GFP::*LGG-1* and P-rsks-1 levels should be quantified.

2. The authors suggest that C22 GlcCer regulates clathrin membrane localization and downstream SKN-1 activation in germline-deficient *C. elegans*. To prove this, effect of C22 ceramide supplement on CHC-1 membrane localization should be tested in *elo-3* RNAi, *sptl-1* RNAi and *cgt-3* RNAi worms.

3. To prove that the cDCF-positive granules are autolysosomes as concluded by the authors, additional experiments are needed.

1) In Fig. 6D and S6F, the image quality of GFP::*LGG-1* and NUC-1::*mCherry* is not sufficient. It is difficult to see whether the two fluorescent proteins are colocalized or adjacent. GFP fluorescence is normally quenched and thus invisible in the acidic compartments such as lysosome. It is strange that GFP::*LGG-1* can be seen in lysosomes, especially in *elo-3* RNAi, *sptl-1* RNAi and *cgt-3* RNAi worms in which lysosomal acidity is increased.

2) EM analyses are needed to determine whether enlarged cDCF-positive granules are autolysosomes.

4. The authors suggest that C22 GlcCer plays a crucial role in lysosome homeostasis, but the "lysosome homeostasis" is not clearly defined in this study. The authors stated that "C22 GlcCer and clathrin affect lysosome activity and morphology." Is lysosome activity increased with reduced C22 GlcCer production and knockdown of clathrin? If so, why lysosome size is enlarged? Is the lysosome enlargement due to defective cargo digestion or reformation of lysosomes as implied by the authors? This should be clarified and tested.

5. The authors may examine ALR process directly to see whether it is affected by reduced C22 GlcCer or in clathrin-defective mutants. This process has been characterized previously in *C. elegans* (Gan et al., JCB 2019 Vol. 218 No. 8).

6. Cholesterol supplementation rescued CHC-1::*GFP* localization and reversed the lysosome phenotype in animals with reduced C22 GlcCer (Fig. 5E, S6E). How does CHC-1, which is localized to the apical cell membrane, regulate lysosome morphology and activity? Does CHC-1 regulate lysosome morphology/activity directly? Is it localized to lysosome? If so, is the lysosomal localization affected by reduced C22 GlcCer and reversed by cholesterol supplement?

7. What is the link between enlarged cDCF-positive granules (autolysosomes?) and TOR activation? What exactly is the C22 GlcCer-mediated lysosome homeostasis? Does C22GlcCer promote lysosome degradation and/or reformation? In mammalian cells, mTOR is reactivated after prolonged starvation to promote ALR. Inhibition of mTOR activity blocks ALR. How presence of enlarged autolysosomes (if they indeed are), which may be caused by defective ALR as implied by the authors, leads to TOR activation? This issue should be clarified and tested.

8. The authors examined GFP::*LGG-1* protein levels to indicate autophagy activity and TOR activation, which is not appropriate. To examine autophagy activity, endogenous *LGG-1* levels instead of overexpressed GFP::*LGG-1* should be examined and *LGG-1* processing should be shown.

The P-rsks-1 levels should be quantified, and the total *RSKS-1* level should also be shown, especially in *glp-1*, which has no detectable P-rsks-1. Moreover, P-rsks-1 should be examined in *tor*-defective backgrounds to verify the fidelity of the assay.

9. How is TOR inhibition linked to *SKN-1* activation? This point should be at least discussed.

Minor points:

1. In Fig. 1D, the nuclear occupation of *SKN-1* is not clearly. Enlarged images should be shown. In 1E (and other similar figure panels), the y-axis should indicate % of *SKN-1*::GFP level in the nucleus.
2. Fig. S1B, C: genotype should be labeled.
3. Why C24-ceramide has effect on *gst-4p*::GFP induction in *sptl-1* RNAi but not in *elo-3* RNAi worms?
4. Fig. 5C, *chc-1* RNAi causes significantly shortened lifespan in WT, which may be caused by a general sickness of *chc-1* knockdown but not a specific effect to *glp-1* lifespan. The authors should clarify this point.

Reviewer #2 (Remarks to the Author):

Wang et al. study germline loss induced longevity in *C. elegans*, focusing on the relationship between the transcription factors *SKN-1* and membrane lipids. They show that loss of very long chain FA synthesis and specifically glycosphingolipids shortens life span of longevity, alters membrane localization of clathrin, and leads to abnormal vesicle formation. This appears to activate Tor, thus explaining the short life span.

The study suggests a new mechanism of how *SKN-1* activation in germline-less animals can promote longevity, adding to previous reports on how lipid signals activate *SKN-1* in this context. However, there are issues with the readout used to pinpoint *SKN-1*, and lack of integration of previous knowledge (Tor regulation of *SKN-1* etc). there are also technical issues with the life span and especially the stress response data.

Major

1. The rationale doesn't adequately integrate that vast amount of work that has been done on the role of lipid metabolism in the regulation of life span and specifically that of germ-line less mutants, including via *SKN-1*. The authors state "The function of FA desaturation in germline-mediated longevity has been explored, identifying $\Delta 9$ desaturase *FAT-6* as a regulator of lifespan in *C. elegans* 16,18. However, whether fatty acid elongation is involved in longevity response is still unknown." This is an oversimplification omitting the many studies that revealed diverse effects for lipids and lipid metabolism in *C. elegans* longevity; including paper such as PMIDs: 25474470, 31408455, 31379987, 31315038, 30911178, 30713071, 30300667, 30269951, 28772063, 28379943, and many more. Effects of sphingolipids have been studied in PMID 30184537, 25437839, and 23894595. Further, the regulation of FA elongases in germline-less animals was noted in PMID 26196144 and 25474470. The former paper by Steinbaugh et al. in fact identified a role for fatty acid regulation of *SKN-1* in germline-less longevity. The fact that the authors could not reproduce these published effects of oleic acid on *SKN-1* activity (PMID: 26196144) is

concerning and should be discussed.

2. Life span and stress response data are inadequate, and must be expanded. First, it is indicated "at least two independent experiments" were done. 3 would be better and must be done for all key experiments (e.g. life span and stress assays in main figures). Second, the stress response assays are done with very low n of 30-50 animals; this is inadequate and more in line with a pilot experiment (each worm represents 2-3% of total pop. survival measured!). At least 80-100 should be used. Third, the data are only shown in Table S1-2 for one replicate; this is insufficient, the data for all replicates must be reported to assess variation. Table should also Fourth, the authors state that *elo-3* "dramatically" shortens LSPN in *glp-1* mutants (looks like ~ 25 to 16 days) but only modestly in *daf-2* and *eat-2* mutants (28 to 22 days, and 25 to 21 days); these are all significant (not shown in Fig., please add), so please use a more quantitative and nuanced description and drop adjectives to describe the actual data.

3. Fatty acid supplementation: The authors should use their lipid measurement techniques to ascertain that C21:0 and C22:0 are taken up efficiently by worms; otherwise, C22:0's effects might be indirect, perhaps by altering *E. Coli* metabolism.

4. The authors conclude that *elo-3*, BA, and ceramides are sufficient to regulate SKN-1, but this conclusion is not supported by their data. First, the effect of BA supplementation on *gst-4::gfp* expression in Fig 2F and of C22 cer in Fig S3I is unconvincing; please quantify. Second, using just one readout is insufficient; more bona fide SKN-1 target genes must be studied by qPCR or, better, RNA-seq. I really need to emphasize that the repeated use of *gst-4* alone as proxy for SKN-1 activity is unacceptable; this gene is also regulated by *nhr-49*, which is also critical for *glp-1* longevity and gene activation (PMIDs: 30297383, 25474470). Third, for the survival assays in Figs 2 and 3 and S2 and S3, the authors need to perform BA supplementation with control and *skn-1* RNAi or with Wt and *skn-1* mutants; this alone establishes *skn-1* as the effector. Fourth, the conclusion "...the enhanced oxidative stress resistance and longevity of *glp-1* mutants were largely abolished by *sptl-1* RNAi" is wrong – the fractional effect appears very similar (50%) in all conditions, hence loss of *sptl-1* simply makes animals of any genotypes short lived. One tool that could be really valuable in these experiments would be the *skn-1gf* alleles; these might be sufficient to overcome the defects of reduced BA and ceramide levels.

5. All figures: Please replace all bar graphs with dot plots (especially for e.g. qPCR in Fig 1 etc); consult PMID: 32346721 for ideal presentation that allows visualization of individual repeats.

6. The analyses on lysosome structure and function should be expanded. All of Fig 6 lacks quantification of number and size of granules; this should be done to best capture the effects. It would also be good to support studies on autophagy with more than just LGG-1 readouts.

7. Whenever showing negative RNAi data such as on *cgt-1* and *cgt-2*, the authors need to ascertain that these genes were actually KD-ed efficiently; otherwise, simply lack of effective RNAi could cause lack of effects (along those lines, Fig S4C misses the *cgt-3* alone condition).

8. The well described role that the Tor pathway plays in SKN-1 activation in longevity and stress responses is not well appreciated. The key paper from the Blackwell lab is cited, but the data not adequately discussed.

9. Statistics & data analysis : Fig 1B, C, 2A, S2A, B, etc: control samples need error bars; Fig 1G-H, S1D-E, S2E, etc need statistics in the figure.

Minor

1. Add line numbers for all submissions in future

2. Fig 1C: regulation of *elo-3* in *glp-1* is inconsistent with published data in PMID 26196144 and 25474470. Explain.

3. Fig 1D: is this loss of nuclear localization, or loss of expression? Please use immunoblot to show protein is still expressed or else explain the limitation of the assay. Otherwise, conclusion P4 "completely abrogated the SKN-1 nuclear accumulation" not adequately qualified or supported by the data.

4. Fig 1H: data inconsistent with previous report showing short lifespan on *elo-3* RNAi. Explain.

5. Fig S3C lacks a control

Grammar & typos:

1. P1: "...changes is causally"

2. P2: "...One the membrane, sphingolipids..."

3. P2: authors should cite PMID 25474470 when discussing lipid changes in *glp-1* long lived worms

4. Figure S4 title: "germline loss"

5. P12: "To provide clue"
6. P12: "sharing many common feathers with"
7. P13: "gremlin"

Reviewer #3 (Remarks to the Author):

The article by Wang et al. describes the role of a glycosphingolipid in lysosome homeostasis. More specifically the authors show that the elongase ELO-3 is responsible for the biosynthesis of behenic acid and other saturated long chain fatty acids. Extensive work shows that specifically C22-Glucosylceramide is required for the correct membrane localization of clathrin, which in turn is required for autophagic lysosome reformation.

The article is well-written and easy to follow, however I am missing a few specific points.

1. The authors show reduction of C21:0, C22:0, C23:0 and C24:0 fatty acids upon *elo-3* RNAi. However, levels are not completely abolished, so I am wondering if a similar reduction is seen in the sphingolipids and more specifically, if specific subclasses of sphingolipids show a reduction. Analysis of intact sphingolipids would shed some light on how direct the effect is and if levels of fatty acid are dropping generally in all lipids or only specific classes. Furthermore, the authors used an acidic transesterification, but the amide bond in sphingolipids is stable under these conditions. Use of 2 M KOH is required to analyze fatty acids bound in sphingolipids. Since the authors performed lipid fractionation, the sphingolipid fraction could be used for such analysis.
2. Analysis of intact lipid analysis would additionally show if other sphingolipid species are also changed and to which extent. Based on the feeding experiments it seems that C22-ceramide is the main sphingolipid important, but intact lipid analysis would reveal the general impact of *elo-3* reduction on sphingolipids.
3. Worms were supplemented with ceramides. Which sphingoid based did these ceramides contain? C18 or C17iso? Since C17iso sphingolipids are not commercially available I would guess C18. Please additionally state the exact lipid name. If it was C17iso, the source or synthesis, etc. should be added to the manuscript. Was also a direct supplementation with glycosphingolipids tested=
4. Glycosphingolipids in *C. elegans* contain 2-hydroxy fatty acids bound as N-acyls. For example, Chitwood et al. [1] showed that 31.6 % are 22:0(OH). Zhu et al. showed that a sphingolipid-TORC1 pathway is required for postembryonic development. Different mutants and supplements have been tested, which included *fath-1*, which is adding the 2-hydroxyl group to fatty acids. RNAi of *fath-1* caused larval arrest, indicating that the hydroxyl group is required [2]. I am wondering if *fath-1* was also tested to be required for lysosome homeostasis?

After consideration of raised points and revision, the article is good to be published.

1. Chitwood D, Lusby W, Thompson M, Kochansky J, Howarth O. The glycosylceramides of the nematode *Caenorhabditis elegans* contain an unusual, branched-chain sphingoid base. *Lipids*. 1995;30(6):567-73. doi: 10.1007/BF02537032.
2. Zhu H, Shen H, Sewell AK, Kniazeva M, Han M. A novel sphingolipid-TORC1 pathway critically promotes postembryonic development in *Caenorhabditis elegans*. *Ahringer J*, editor 2013 2013-05-21 16:07:10.

We thank the reviewers for their professional and insightful comments. According to their suggestions, we have performed additional experiments and edited the text to improve our manuscript.

Reviewer #1 (Remarks to the Author):

In this study, the authors identified C22 GlcCer as a key molecule involved in SKN-1 activation and lifespan extension in germline-deficient C. elegans. The authors showed that C22 GlcCer is required for membrane localization of clathrin. Defects in C22 GlcCer production causes enlarged autolysosomes. Moreover, TOR activity seems to increase in elo-3 RNAi worms and inhibiting TOR activity restored SKN-1 reporter gene expression, TBHP resistance and lifespan in elo-3 RNAi worms. The authors proposed that C22 GlcCer affects membrane protein clathrin and thus modulates lysosome homeostasis, TOR signaling and SKN-1 activation.

This manuscript provided clear data that C22 GlcCer production is important for SKN-1 activation, TBHP resistance and lifespan extension in germline-deficient glp-1 mutant. However, the mechanistic link between C22 GlcCer, ALR and TOR activity is not established convincingly.

Major comments:

1. For most cellular and molecular assays, data were compared but not quantified.

1) As the key assay for SKN-1 transcriptional activity utilized throughout the whole study, the gst-4p::GFP intensity should be quantified and then compared in different strains/conditions to clearly indicate changes in SKN-1 activity.

Response: We thank the reviewer for this important suggestion. The GFP intensity has been quantified with statistical analysis in the revised manuscript.

2) In Fig. 6 and S6, cDCF intensity and lysosome size should be quantified.

Response: We have quantified the cDCF intensity and size as the reviewer suggested.

3) In Fig 7A and B, GFP::LGG-1 and P-rsks-1 levels should be quantified.

Response: As the reviewer mentioned in point #8, the GFP::LGG-1 data is not quite suitable for

measuring the autophagic activity, we have replaced it with two new assays (EM analysis and a autophagosome fluorescent maker) with quantitative analysis (please see our response to point #8). P-RSKS-1 levels are also quantified (Fig. 7b), which consist with the conclusion.

2. The authors suggest that C22 GlcCer regulates clathrin membrane localization and downstream SKN-1 activation in germline-deficient C. elegans. To prove this, effect of C22 ceramide supplement on CHC-1 membrane localization should be tested in elo-3 RNAi, sptl-1 RNAi and cgt-3 RNAi worms.

Response: We have performed the experiments as the reviewer suggested and provided the results showing C22 ceramide addition restored the CHC-1 membrane localization in animals treated with *elo-3* RNAi, *sptl-1* RNAi, or *cgt-3* RNAi (Fig. 5d and S5c).

3. To prove that the cDCF-positive granules are autolysosomes as concluded by the authors, additional experiments are needed.

1) In Fig. 6D and S6F, the image quality of GFP::LGG-1 and NUC-1::mCherry is not sufficient. It is difficult to see whether the two fluorescent proteins are colocalized or adjacent. GFP fluorescence is normally quenched and thus invisible in the acidic compartments such as lysosome. It is strange that GFP::LGG-1 can be seen in lysosomes, especially in elo-3 RNAi, sptl-1 RNAi and cgt-3 RNAi worms in which lysosomal acidity is increased.

2) EM analyses are needed to determine whether enlarged cDCF-positive granules are autolysosomes.

Response: We thank the reviewer for this insightful comment. To prove that the reduction of C22 GlcCer results in the accumulation of enlarged autolysosomes (ALs), multiple additional experiments were performed: (1) GFP::LGG-1 and NUC-1::mCherry co-localization experiments were re-performed and much clearer microscopy images were obtained to show that these two fluorescent markers were indeed colocalized (Fig. 6d). (2) We examined another autophagosome (AP) marker GFP::LGG-2 and found it also colocalized with NUC-1::mCherry in the enlarged vesicles in *elo-3/sptl-1/cgt-3* RNAi worms (Fig. S6j). (3) We employed mCherry::GFP::LGG-1, a reliable fluorescent indicator in *C. elegans* that distinguishes between APs (GFP + mCherry) and ALs

(mCherry only due to lysosomal GFP quenching) [*eLife* 2017, 6:e18459]. The data showed that worms with reduced C22 GlcCer had significantly enlarged ALs, while the ALs were relatively smaller in the control worms (Fig. 6e). (4) We further performed TEM analysis to consolidate the conclusion as the reviewer suggested. As expected, reduced C22 GlcCer production led to the accumulation of enlarged ALs (Fig. 6f). These new data, together with our previous finding that knockdowns of autophagic genes suppressed the enlargement of cDCF vesicles (Fig. S6k), suggest that reduction of C22 GlcCer indeed results in accumulation of enlarged ALs.

Currently, we do not know why GFP::LGG-1 and GFP::LGG-2 were not quenched in these enlarged ALs. Intriguingly, in the JCB paper mentioned by the reviewer (point #5, *JCB* 2019, 218:2619-2637), the enlarged ALs in *slc-36.1* mutants are also positive for GFP::LGG-2, suggesting it may be a general phenotype. Also, we notice that only certain GFP were quenched, as the GFP signals from mCherry::GFP::LGG-1 are absent in the enlarged ALs of *elo-3/sptl-1/cgt-3* RNAi worms (Fig. 6e). Indeed, we observed that the GFP signals between GFP::LGG-1 and mCherry::GFP::LGG-1 are completely different (Fig. 6d versus Fig. 6e), although they both reflect LGG-1 localization. Our speculation is that the fused mCherry::GFP may be more sensitive to quenching than GFP alone in the enlarged ALs. We have added this point to the revised manuscript (Page 15, Line 331-334).

4. The authors suggest that C22 GlcCer plays a crucial role in lysosome homeostasis, but the “lysosome homeostasis” is not clearly defined in this study. The authors stated that “C22 GlcCer and clathrin affect lysosome activity and morphology.” Is lysosome activity increased with reduced C22 GlcCer production and knockdown of clathrin? If so, why lysosome size is enlarged? Is the lysosome enlargement due to defective cargo digestion or reformation of lysosomes as implied by the authors? This should be clarified and tested.

Response: Please see our response to point #5.

5. The authors may examine ALR process directly to see whether it is affected by reduced C22 GlcCer or in clathrin-defective mutants. This process has been characterized previously in C. elegans (Gan et al., JCB 2019 Vol. 218 No. 8).

Response: We thank the reviewer for these professional concerns. As the reviewer mentioned in

point #4, the enlarged ALs can be due to defective lysosomal digestion or impaired ALR. We firstly performed two independent assays to examine the lysosomal activity: the cathepsin activity assay and NUC-1::mCherry cleavage assay. The results showed that the lysosomal activities in animals with reduced C22 GlcCer production or knockdown of clathrin were either normal or even enhanced (Fig. S6h and S6i), suggesting that the enlarged ALs are not due to defective lysosomal digestion.

We next examined the ALR process directly as the reviewer suggested. (1) We first examined the NUC-1::mCherry lysosome dynamics and found that the lysosome reformation from ALs could be frequently observed in the control worms, but was barely seen in *elo-3/sptl-1/cgt-3* RNAi worms (Fig. 6g). (2) We further challenged the animals with starvation, a condition known to promote ALR. In the control animals, we observed lots of tubules extending from the NUC-1::mCherry lysosomes, likely indicating active ALR. However, this phenotype was completely abrogated by reduction of C22 GlcCer (Fig. 6h). Altogether, these data suggest that the enlarged ALs in animals with reduced C22 GlcCer production result from impaired ALR.

6. Cholesterol supplementation rescued CHC-1::GFP localization and reversed the lysosome phenotype in animals with reduced C22 GlcCer (Fig. 5E, S6E). How does CHC-1, which is localized to the apical cell membrane, regulate lysosome morphology and activity? Does CHC-1 regulate lysosome morphology/activity directly? Is it localized to lysosome? If so, is the lysosomal localization affected by reduced C22 GlcCer and reversed by cholesterol supplement?

Response: We thank reviewer for this insightful concern. To examine whether CHC-1 is localized on the lysosomal membrane, we stained the lysosomes with lyso tracker red and examined its colocalization with CHC-1::GFP. The results showed only a few lysosomes were positive for weak CHC-1::GFP signals on the membranes (Fig. S6g), which made it difficult to further explore the regulation of CHC-1::GFP on the lysosomes. Since the mammalian clathrin was found to directly regulate ALR, we propose that clathrin on the lysosomes may also be regulated by C22 GlcCer, which in turn affects lysosome homeostasis.

7. What is the link between enlarged cDCF-positive granules (autolysosomes?) and TOR activation? What exactly is the C22 GlcCer-mediated lysosome homeostasis? Does C22GlcCer promote

lysosome degradation and/or reformation? In mammalian cells, mTOR is reactivated after prolonged starvation to promote ALR. Inhibition of mTOR activity blocks ALR. How presence of enlarged autolysosomes (if they indeed are), which may be caused by defective ALR as implied by the authors, leads to TOR activation? This issue should be clarified and tested.

Response: We thank the reviewer for the important concern and apologize for not stating this point clearly. As we proved above, reduced C22 GlcCer production leads to enlarged ALs due to defective ALR. Mammalian mTOR is activated after prolonged starvation to promote ALR and terminate autophagy to restore the lysosome homeostasis. We speculated that impaired ALR and/or ALs accumulation may tell the cell that it needs to activate ALR (through mTOR induction) and maintain lysosome homeostasis. Therefore, in our context, numerous enlarged ALs may continuously signal the cell to activate TOR.

To test whether AL accumulation could indeed activate TOR, we knocked down ALR genes *slc-36.1* and *ppk-3*, both of which led to the accumulation of enlarged ALs (*JCB* 2019, 218:2619-2637, and Fig. S7b), and observed similar activation of TOR (Fig. 7a and 7b), suggesting that AL accumulation could induce TOR activation in *C. elegans*. We have carefully edited the text to make the logic much clearer.

8. The authors examined GFP::LGG-1 protein levels to indicate autophagy activity and TOR activation, which is not appropriate. To examine autophagy activity, endogenous LGG-1 levels instead of overexpressed GFP::LGG-1 should be examined and LGG-1 processing should be shown. The P-rsks-1 levels should be quantified, and the total RSKS-1 level should also be shown, especially in glp-1, which has no detectable P-rsks-1. Moreover, P-rsks-1 should be examined in tor-defective backgrounds to verify the fidelity of the assay.

Response: We agree with the reviewer for this concern. Endogenous LGG-1 processing is more suitable for measuring autophagy activity, but cannot distinguish between APs and ALs. In our model, we propose that accumulation of ALs may activate TOR, which could prevent AP formation in animals with reduced C22 GlcCer. Therefore, we believe the amount of APs, instead of overall autophagic activity (including APs and ALs), is more suitable for measuring the TOR activity in our system. Two independent assays were used: mCherry::LGG-1::GFP reporter (APs positive for both

red and green signals) and TEM analysis. Both experiments showed that the numbers of APs were indeed decreased in animals with reduced C22 GlcCer (Fig. 7a and S7a), consistent with an enhanced TOR activity.

For the measurement of total RSKS-1, we tried two antibodies (Cell signaling, #2708 and #9202), but unfortunately could not detect any clear signals for the *C. elegans* RSKS-1. This may be a general problem for the *C. elegans* studies as we noticed similar situations in other *C. elegans* papers, which use tubulin or actin instead of the total RSKS-1 (*Nature* 2017, 541:102-106; *eLife* 2017, 6:e31268). As the reviewer suggested, we included the TOR pathway defective controls (*let-363/TOR* RNAi and *ragc-1/RagC* RNAi) to verify the indicated p-RSKS-1 band is regulated by TOR. The western blot data were also quantified (Fig. 7b).

Together, these data suggest that reduced C22 GlcCer production could activate TOR.

9. How is TOR inhibition linked to SKN-1 activation? This point should be at least discussed.

Response: Regulation of SKN-1 by TOR pathway has been well recognized and the underlying mechanisms have begun to be uncovered. We have added the reference before the TOR section. (Page 16, Line 352-355).

Minor points:

1. In Fig. 1D, the nuclear occupation of SKN-1 is not clearly. Enlarged images should be shown. In 1E (and other similar figure panels), the y-axis should indicate % of SKN-1::GFP level in the nucleus.

Response: We thank the reviewer for the suggestion and have changed the figures as suggested.

2. Fig. S1B, C: genotype should be labeled.

Response: We thank the reviewer for the suggestion and the genotypes have been added.

*3. Why C24-ceramide has effect on *gst-4p::GFP* induction in *sptl-1* RNAi but not in *elo-3* RNAi worms?*

Response: We thank the reviewer for this point. As the reviewer #3 suggested, we quantified sphingolipids species in detail and found that very long-chain (C>20) ceramides were reduced in *elo-3* RNAi worms as expected (Fig. S4b). However, a compensatory increase of C18 ceramide was also observed (Fig. S4b). Since it has been reported that ceramides with different acyl chains could function oppositely (e.g. *Science* 2009, 324:381-384), we speculate that C18 ceramide may counteract the effects of C24 ceramide as shown by the expression of *gst-4p::GFP*. C24 ceramide, due to its structural similarity to C22 ceramide, could also restore the *gst-4p::GFP* activity in *sptl-1* RNAi animals, which reduce the contents of all ceramides regardless of the length of acyl chains. While in *elo-3* RNAi worms, the increased C18 ceramide may disturb the effects of C24 ceramide. We have added this to the discussion section (Page 18, Line 407-411).

4. Fig. 5C, *chc-1* RNAi causes significantly shortened lifespan in WT, which may be caused by a general sickness of *chc-1* knockdown but not a specific effect to *glp-1* lifespan. The authors should clarify this point.

Response: We have edited the text to clarify this point (Page 11, Line 253-255). The new sentence reads: *It should be noted that chc-1 RNAi dramatically shortened lifespan of wild-type animals, therefore it cannot be ruled out that the effects of chc-1 RNAi on lifespan may be due to general sickness.* The explanation of the lifespan data does not affect the conclusion that clathrin is a regulator of SKN-1 activation in germline-deficient *C. elegans*.

Reviewer #2 (Remarks to the Author):

Wang et al. study germline loss induced longevity in C. elegans, focusing on the relationship between the transcription factors SKN-1 and membrane lipids. They show that loss of very long chain FA synthesis and specifically glycosphingolipids shortens life span of longevity, alters membrane localization of clathrin, and leads to abnormal vesicle formation. This appears to activate Tor, thus explaining the short life span.

The study suggests a new mechanism of how SKN-1 activation in germline-less animals can promote longevity, adding to previous reports on how lipid signals activate SKN-1 in this context. However,

there are issues with the readout used to pinpoint SKN-1, and lack of integration of previous knowledge (Tor regulation of SKN-1 etc). there are also technical issues with the life span and especially the stress response data.

Major

*1. The rationale doesn't adequately integrate that vast amount of work that has been done on the role of lipid metabolism in the regulation of life span and specifically that of germ-line less mutants, including via SKN-1. The authors state "The function of FA desaturation in germline-mediated longevity has been explored, identifying $\Delta 9$ desaturase FAT-6 as a regulator of lifespan in *C. elegans* 16,18. However, whether fatty acid elongation is involved in longevity response is still unknown." This is an oversimplification omitting the many studies that revealed diverse effects for lipids and lipid metabolism in *C. elegans* longevity; including paper such as PMIDs: 25474470, 31408455, 31379987, 31315038, 30911178, 30713071, 30300667, 30269951, 28772063, 28379943, and many more. Effects of sphingolipids have been studied in PMID 30184537, 25437839, and 23894595. Further, the regulation of FA elongases in germline-less animals was noted in PMID 26196144 and 25474470. The former paper by Steinbaugh et al. in fact identified a role for fatty acid regulation of SKN-1 in germline-less longevity. The fact that the authors could not reproduce these published effects of oleic acid on SKN-1 activity (PMID: 26196144) is concerning and should be discussed.*

Response: We thank the reviewer for this important concern. We have added the references as the reviewer mentioned, but due to the limitation of reference numbers (not exceed 70), we only included those that are closely related to our work in the revised manuscript:

(1) We first expanded the introduction section by adding a new paragraph introducing the link between fatty acid metabolism and aging (Page 2-3, Line 44-50).

(2) As for the role of fatty acid metabolism in germline-deficient worms, we have added several references introducing the involvement of fatty acid metabolic enzymes in germline-mediated longevity response (Page 3, Line 59-62).

(3) The regulations of fatty acid elongases expression in germline-deficient worms have been included in the introduction (Page 3, Line 68-69)

(4) For the rational of sphingolipids, references connecting sphingolipids and aging were cited at the

beginning of the sphingolipid section (Results) (Page 7, Line 147-150).

For the role of oleic acid (OA), our data showed that BA (Fig. 2d), but not OA (Fig. S2e), could restore the *gst-4p::GFP* activity in *elo-3* RNAi worms, suggesting ELO-3 controls SKN-1 activity through BA production. OA could regulate SKN-1 as previous papers reported, but it does not function in ELO-3 pathway. We think BA and OA may function in parallel or converge on a downstream point to activate SKN-1 in the germline loss worms, which needs further exploration. After the OA data, we added a sentence to clarify this point: ..., suggesting ELO-3 does not regulate SKN-1 via OA production (Page 6, Line 135-136).

*2. Life span and stress response data are inadequate, and must be expanded. First, it is indicated "at least two independent experiments" were done. 3 would be better and must be done for all key experiments (e.g. life span and stress assays in main figures). Second, the stress response assays are done with very low n of 30-50 animals; this is inadequate and more in line with a pilot experiment (each worm represents 2-3% of total pop. survival measured!). At least 80-100 should be used. Third, the data are only shown in Table S1-2 for one replicate; this is insufficient, the data for all replicates must be reported to assess variation. Table should also Fourth, the authors state that *elo-3* "dramatically" shortens LSPN in *glp-1* mutants (looks like ~ 25 to 16 days) but only modestly in *daf-2* and *eat-2* mutants (28 to 22 days, and 25 to 21 days); these are all significant (not shown in Fig., please add), so please use a more quantitative and nuanced description and drop adjectives to describe the actually data.*

Response: We thank the reviewer for these important concerns. As the review suggested, we have re-performed all stress resistance assays with an average n number of 80-100 and with three repeats for main figures. All replicates data have been included in Table S1 and S2.

The lifespan data from *elo-3* RNAi animals have been described more accurately as the reviewer suggested (Page 5, Line 109-110). The new sentence reads: ...knockdown of *elo-3* also suppressed the lifespan of *daf-2* and *eat-2* mutants, although to a lesser extent compared with the germline-deficient animals.

3. *Fatty acid supplementation: The authors should use their lipid measurement techniques to ascertain that C21:0 and C22:0 are taken up efficiently by worms; otherwise, C22:0's effects might be indirect, perhaps by altering E. Coli metabolism.*

Response: We thank the reviewer for this valuable suggestion. We have examined the fatty acid contents and the new data showed that both C21:0 and C22:0 have been efficiently taken up, as the corresponding fatty acids levels were completely restored in *elo-3* RNAi worms (Fig. S2c).

4. *The authors conclude that *elo-3*, BA, and ceramides are sufficient to regulate SKN-1, but this conclusion is not supported by their data. First, the effect of BA supplementation on *gst-4::gfp* expression in Fig 2F and of C22 cer in Fig S3I is unconvincing; please quantify. Second, using just one readout is insufficient; more bona fide SKN-1 target genes must be studied by qPCR or, better, RNA-seq. I really need to emphasize that the repeated use of *gst-4* alone as proxy for SKN-1 activity is unacceptable; this gene is also regulated by *nhr-49*, which is also critical for *glp-1* longevity and gene activation (PMIDs: 30297383, 25474470). Third, for the survival assays in Figs 2 and 3 and S2 and S3, the authors need to perform BA supplementation with control and *skn-1* RNAi or with *Wt* and *skn-1* mutants; this alone establishes *skn-1* as the effector. Fourth, the conclusion "...the enhanced oxidative stress resistance and longevity of *glp-1* mutants were largely abolished by *sptl-1* RNAi" is wrong – the fractional effect appears very similar (50%) in all conditions, hence loss of *sptl-1* simply makes animals of any genotypes short lived. One tool that could be really valuable in these experiments would be the *skn-1gf* alleles; these might be sufficient to overcome the defects of reduced BA and ceramide levels.*

Response: We thank the reviewer for raising these important concerns. We performed additional experiments to consolidate our conclusion on SKN-1 regulation:

1. *gst-4p::GFP* has been quantified to show that BA and C22 ceramide are sufficient to induce its expression (Fig. 2f and S3i).
2. qPCR assays were performed to confirm that BA and C22 ceramide can induce the expression of other SKN-1 target genes (Fi. S2f and S3j). We also measured SKN-1 target genes expression for other key experiments throughout the manuscript, including the *sptl-1* and *cgt-3* RNAi (Fig. 3b and 4c). In addition, SKN-1::GFP nuclear occupancy assays were performed for several major

data, including the *cgt-3* RNAi (Fig. 4d), cholesterol supplementation (Fig. 5g and S5f) and RNAi of TOR components (Fig. 7h and S7h). These new data all support our original conclusions.

3. As the reviewer suggested, we tested the effects of BA and C22 ceramide in *skn-1* RNAi animals and found that these lipids promoted oxidative stress resistance in a SKN-1-dependent manner (Fig. 2g and S3k), further suggesting the involvement of SKN-1 in BA/C22 ceramide effects.

For *sptl-1* RNAi lifespan data, we agree that *sptl-1* RNAi could cause general sickness. As *sptl-1* RNAi theoretically decreases the contents of all sphingolipid species, it is reasonable to observe the general sickness. We also performed the lifespan assay using *skn-1* gain of function (*skn-1* gf) mutants as the reviewer suggested, however, *skn-1* gf mutation unexpectedly **shortened** the lifespan of control germline-deficient *glp-1* worms group (See figure below), therefore we did not observe any reversing effects of *skn-1* gf mutation on the lifespan of *sptl-1* RNAi-treated *glp-1* mutants (See figure below).

We added a sentence to explain the *sptl-1* RNAi lifespan data, which reads as: *It should be noted that the lifespan of wild-type animals was also reduced significantly by sptl-1 RNAi, suggesting that global inhibition of sphingolipid production may cause general sickness* (Page 7, Line 158-161). Although this experiment alone may have multiple explanations, the conclusion of this part does not change as the other data (*sptl-1* RNAi suppresses SKN-1 activation while C22 ceramide could reverse its effects) strongly suggest that specific sphingolipid species accounts for SKN-1 activation in *glp-1* mutants.

5. All figures: Please replace all bar graphs with dot plots (especially for e.g. qPCR in Fig 1 etc); consult PMID: 32346721 for ideal presentation that allows visualization of individual repeats.

Response: We have replaced the bar graphs with scattered plots in revised manuscript.

6. The analyses on lysosome structure and function should be expanded. All of Fig 6 lacks quantification of number and size of granules; this should be done to best capture the effects. It would also be good to support studies on autophagy with more than just LGG-1 readouts.

Response: The size and intensity of cDCF granules have been quantified. Moreover, this part of manuscript has been substantially revised as the reviewer #1 suggested (Please see our responses to point #3-7 of the reviewer #1).

Autophagy has been reevaluated by two independent assays as detailed in the response to point #8 of the reviewer #1. The new data support the suppression of autophagosome formation by *elo-3* RNAi (Fig 7a and S7a), which is consistent with the induction of TOR.

*7. Whenever showing negative RNAi data such as on *cgt-1* and *cgt-2*, the authors need to ascertain that these genes were actually KD-ed efficiently; otherwise, simply lack of effective RNAi could cause lack of effects (along those lines, Fig S4C misses the *cgt-3* alone condition).*

Response: Thank you for this suggestion. We have confirmed the knocking down effects of *cgt-1* and *cgt-2* RNAi (Fig. S4e) and reperformed the experiment including *cgt-3* RNAi alone condition (Fig. S4d).

8. The well described role that the Tor pathway plays in SKN-1 activation in longevity and stress responses is not well appreciated. The key paper from the Blackwell lab is cited, but the data not adequately discussed.

Response: We have added several sentences to clearly introduce the roles of TOR components in SKN-1 regulation (Page 16, Line 352-355).

9. Statistics & data analysis : Fig 1B, C, 2A, S2A, B, etc: control samples need error bars; Fig 1G-H, S1D-E, S2E, etc need statistics in the figure.

Response: In each experiment, data were normalized to controls, therefore each control value is 1

and the corresponding SD value was 0. This makes the error bars invisible. Statistics have been added to lifespan figures as the reviewer suggested.

Minor

1. Add line numbers for all submissions in future.

Response: Thank you for suggestion. Line numbers have been added.

2. Fig 1C: regulation of *elo-3* in *glp-1* is inconsistent with published data in PMID 26196144 and 25474470. Explain.

Response: For paper PMID 25474470, we have not found the data on *elo-3* regulation. In the Blackwell lab paper (PMID 26196144), RNA-seq data showed a 1.5-fold increase in *elo-3* mRNA levels in *glp-1* mutants (Fig 5A in their paper). Our data showed that *elo-3* mRNA levels were 3-fold higher in *glp-1* mutants (Fig. 1c). The conclusions are consistent, while the differences in fold change may be due to different methods used (RNA-sequencing versus qPCR).

3. Fig 1D: is this loss of nuclear localization, or loss of expression? Please use immunoblot to show protein is still expressed or else explain the limitation of the assay. Otherwise, conclusion P4 "completely abrogated the SKN-1 nuclear accumulation" not adequately qualified or supported by the data.

Response: We have performed western blot experiment and found that the protein levels of SKN-1::GFP were not affected by *elo-3* RNAi (Fig. S1b), suggesting that SKN-1 nuclear accumulation is indeed abrogated.

4. Fig 1H: data inconsistent with previous report showing short lifespan on *elo-3* RNAi. Explain.

Response: We may miss some reference but can not find previous paper reporting *elo-3* RNAi lifespan data.

5. Fig S3C lacks a control.

Response: We may not understand what the reviewer meant, as Fig. S3c showed the effects of *sptl-1* RNAi on lifespan in *glp-1* mutants, which included both wild-type animals and control RNAi as controls.

Grammar & typos:

1. P1: "...changes is causally"
2. P2: "...One the membrane, sphingolipids..."
3. P2: authors should cite PMID 25474470 when discussing lipid changes in *glp-1* long lived worms
4. Figure S4 title: "germline loss"
5. P12: "To provide clue"
6. P12: "sharing many common feathers with"
7. P13: "gremlin"

Response: We thank the reviewer for pointing out these errors and apologize for our negligence. These errors have been corrected.

Reviewer #3 (Remarks to the Author):

The article by Wang et al. describes the role of a glycosphingolipid in lysosome homeostasis. More specifically the authors show that the elongase ELO-3 is responsible for the biosynthesis of behenic acid and other saturated long chain fatty acids. Extensive work shows that specifically C22-Glucosylceramide is required for the correct membrane localization of clathrin, which in turn is required for autophagic lysosome reformation.

The article is well-written and easy to follow, however I am missing a few specific points.

1. *The authors show reduction of C21:0, C22:0, C23:0 and C24:0 fatty acids upon *elo-3* RNAi. However, levels are not completely abolished, so I am wondering if a similar reduction is seen in the sphingolipids and more specifically, if specific subclasses of sphingolipids show a reduction. Analysis of intact sphingolipids would shed some light on how direct the effect is and if levels of fatty acid are dropping generally in all lipids or only specific classes. Furthermore, the authors used an acidic transesterification, but the amide bond in sphingolipids is stable under these conditions. Use*

of 2 M KOH is required to analyze fatty acids bound in sphingolipids. Since the authors performed lipid fractionation, the sphingolipid fraction could be used for such analysis.

Response: We thank the reviewer for this insightful concern. In the revised manuscript, we collaborated with Dr. Huanhu Zhu, an expert in *C. elegans* sphingolipid study, to quantify the sphingolipid classes in *elo-3* RNAi worms. This time, we extracted sphingolipid species and directly analyzed them using UPLC/QTRAP-MS. The results showed that *elo-3* RNAi reduced very long-chain fatty acids in all major classes of sphingolipids, including ceramide, sphingomyelin and glucosylceramide (Fig. 4a, S4b and S4c). This is consistent with a general role for ELO-3 in fatty acid elongation, and the knockdown of *elo-3* would theoretically affect the corresponding acyl chains of all sphingolipids. Intriguingly, we also found the contents of C18 ceramide and C18 sphingomyelin were substantially elevated (Fig. S4b and S4c), likely due to a compensatory effect for overall sphingolipid maintenance.

*2. Analysis of intact lipid analysis would additionally show if other sphingolipid species are also changed and to which extent. Based on the feeding experiments it seems that C22-ceramide is the main sphingolipid important, but intact lipid analysis would reveal the general impact of *elo-3* reduction on sphingolipids.*

Response: Please see our response to point #1.

3. Worms were supplemented with ceramides. Which sphingoid based did these ceramides contain? C18 or C17iso? Since C17iso sphingolipids are not commercially available I would guess C18. Please additionally state the exact lipid name. If it was C17iso, the source or synthesis, etc. should be added to the manuscript. Was also a direct supplementation with glycosphingolipids tested.

Response: Like the reviewer said, we used commercially available C18 sphingoid base, we have added the lipid name (e.g. C22 ceramide, d18:1/22:0) when we first mentioned it in the revised manuscript (Page 9, Line 185-186).

We did not add GlcCer directly, as GlcCer species with very long-chain fatty acid cannot be dissolved in DMSO, ethanol or water, which renders the supplementation technically impossible.

4. Glycosphingolipids in *C. elegans* contain 2-hydroxy fatty acids bound as N-acyls. For example, Chitwood et al. [1] showed that 31.6 % are 22:0(OH). Zhu et al. showed that a sphingolipid-TORC1 pathway is required for postembryonic development. Different mutants and supplements have been tested, which included *fath-1*, which is adding the 2-hydroxylgroup to fatty acids. RNAi of *fath-1* caused larval arrest, indicating that the hydroxyl group is required [2]. I am wondering if *fath-1* was also tested to be required for lysosome homeostasis?

After consideration of raised points and revision, the article is good to be published.

1. Chitwood D, Lusby W, Thompson M, Kochansky J, Howarth O. The glycosylceramides of the nematode *Caenorhabditis elegans* contain an unusual, branched-chain sphingoid base. *Lipids*. 1995;30(6):567-73. doi: 10.1007/BF02537032.

2. Zhu H, Shen H, Sewell AK, Kniazeva M, Han M. A novel sphingolipid-TORC1 pathway critically promotes postembryonic development in *Caenorhabditis elegans*. *Ahringer J, editor* 2013 2013-05-21 16:07:10.

Response: We thank the reviewer for this important suggestion. We tested *fath-1* RNAi and found that, similar to the *elo-3/sptl-1/cgt-3* RNAi, it suppressed the *gst-4p::GFP* expression (Fig. 4e) and caused the accumulation of enlarged lysosomes (cDCF staining) (Fig. S6d), consistent with the role for glycosphingolipid in SKN-1 regulation and lysosome homeostasis in germline-deficient *C. elegans*.

Reviewer comments, second round –

Reviewer #1 (Remarks to the Author):

The authors have addressed most of my concerns with additional experimental data. My remaining concerns are as follows:

- 1) The GFP::RFP::LC3 assay has been widely used in mammalian cells to indicate autolysosome formation. Here the authors utilized mCHERRY::GFP::LGG-1 assay to distinguish autophagosomes (GFP and mCHERRY) from autolysosomes (mCHERRY only). However, the data are not clear enough as most GFP signal is diffuse in the cytosol (Fig. 6e) and very few autophagosomes are seen (Fig. 6e, 7a). It is probably due to high expression level of the fusion protein. The authors should further verify this assay by checking the mCHERRY::GFP::LGG-1 in *egp-5* mutants which block autophagosome and lysosome fusion. In this case, more GFP- and CHERRY-positive autophagosomes should be observed.
- 2) The authors should examine LGG-1 processing to indicate defects in degradation of autophagic cargo. This assay is not redundant with EM or mCHERRY::GFP::LGG-1 assay, but support the conclusion of late step defects of autophagy process.
- 3) The authors should indicate clearly the cell type and stage where and when the experiments/images were performed/collected in the figures or figure legends. The panel h in figure 6 is not explained in the legend.

Reviewer #2 (Remarks to the Author):

The revised manuscript is much improved. I especially appreciate the much more complete and nuanced discussion of the role of fatty acids in the intro an rationale. I also appreciate the measurement of efficacy of FA supplementation, the expanded and quantitative analysis of SKN-1 activity, and the mRNA quantification of RNAi efficiency.

I am almost satisfied with the additional life span and stress resistance assays that have been completed and with revised Tables S1 -2 include all necessary info. However, for the effect of *elo-3* RNAi on the varying longevity mutants, I would like the authors to include the specific percentage by which this RNAi clone shortens lifespan in each genotype. This will clarify whether the effect is indeed weaker in *daf-2* and *eat-2*, as stated.

Reviewer #3 (Remarks to the Author):

The revised manuscript improved the overall data and story line and in my opinion all necessary corrections and additions have been made. The article is now suitable for publication.

We thank the reviewers for their comments and suggestions. We have modified our manuscript according to their suggestions.

Reviewer #1 (Remarks to the Author):

The authors have addressed most of my concerns with additional experimental data. My remaining concerns are as follows:

*1) The GFP::RFP::LC3 assay has been widely used in mammalian cells to indicate autolysosome formation. Here the authors utilized mCHERRY::GFP::LGG-1 assay to distinguish autophagosomes (GFP and mCHERRY) from autolysosomes (mCHERRY only). However, the data are not clear enough as most GFP signal is diffuse in the cytosol (Fig. 6e) and very few autophagosomes are seen (Fig. 6e, 7a). It is probably due to high expression level of the fusion protein. The authors should further verify this assay by checking the mCHERRY::GFP::LGG-1 in *epg-5* mutants which block autophagosome and lysosome fusion. In this case, more GFP- and CHERRY-positive autophagosomes should be observed.*

Response: We thank the reviewer for this valuable suggestion. We examined the mCHERRY::GFP::LGG-1 in *epg-5* mutants and found that *epg-5* mutation increased the number of autophagosomes in both control and C22 GlcCer-less (*elo-3/sptl-1* RNAi) animals (New Fig. S7b), and partially suppressed the formation of enlarged autolysosomes in *elo-3/sptl-1* RNAi-treated worms (New Fig. S7c) as expected, which verify the previous mCHERRY::GFP::LGG-1 data.

2) The authors should examine LGG-1 processing to indicate defects in degradation of autophagic cargo. This assay is not redundant with EM or mCHERRY::GFP::LGG-1 assay, but support the conclusion of late step defects of autophagy process.

Response: As the reviewer suggested, we examined autophagic degradation of LGG-1::GFP through immunoblot analysis. LGG-1::GFP levels were modestly reduced in animals with less C22 GlcCer, while the cleaved GFP levels, which indicates the autophagic degradation of LGG-1::GFP in autolysosomes, were dramatically suppressed by C22 GlcCer reduction (New Fig. S6l). This data suggests that the enlarged autolysosomes in C22 GlcCer-less animals are associated with impaired degradation activity. The text was edited accordingly (Page 15, Line 343-347).

3) *The authors should indicate clearly the cell type and stage where and when the experiments/images were performed/collected in the figures or figure legends.*

The panel h in figure 6 is not explained in the legend.

Response: We thank the reviewer for this important concern. We have indicated the cell type of each assay in the figures or figure legends. As day 1 adult animals were used throughout the manuscript, this information has been clearly stated in the methods section.

The explanation of Figure 6h has been added to the legend.

Reviewer #2 (Remarks to the Author):

The revised manuscript is much improved. I especially appreciate the much more complete and nuanced discussion of the role of fatty acids in the intro and rationale. I also appreciate the measurement of efficacy of FA supplementation, the expanded and quantitative analysis of SKN-1 activity, and the mRNA quantification of RNAi efficiency.

*I am almost satisfied with the additional life span and stress resistance assays that have been completed and with revised Tables S1 -2 include all necessary info. However, for the effect of *elo-3* RNAi on the varying longevity mutants, I would like the authors to include the specific percentage by which this RNAi clone shortens lifespan in each genotype. This will clarify whether the effect is indeed weaker in *daf-2* and *eat-2*, as stated.*

Response: We thank the reviewer for the comment and suggestion. In the revised manuscript, we have indicated the specific percentage by which *elo-3* RNAi shortens lifespan in each genotype. Now the sentence read as: knockdown of *elo-3* also suppressed the lifespan of *daf-2* (19% reduction) and *eat-2* (15% reduction) mutants, although to a lesser extent compared with the germline-deficient animals (37% reduction) [Page 5, Line 109-111].

Reviewer #3 (Remarks to the Author):

The revised manuscript improved the overall data and story line and in my opinion all necessary corrections and additions have been made. The article is now suitable for publication.

Response: Thank you for the comment.

Reviewer comments, third round –

Reviewer #1 (Remarks to the Author):

The authors have addressed my concerns in the revised manuscript.

Reviewer #2 (Remarks to the Author):

thank you for addressing all concerns.